# SensX: Model-Agnostic Local Feature Attribution via Calibrated Global Sensitivity Analysis

**Manu Aggarwal**                                                      *manu.aggarwal@nih.gov*
*Laboratory of Biological Modeling, NIDDK*
*National Institutes of Health*
*Bethesda, MD, USA*

**Nick Cogan**                                                          *cogan@math.fsu.edu*
*Department of Mathematics*
*Florida State University*
*Tallahassee, FL, USA*

**Vipul Periwal**                                                       *vipulp@niddk.nih.gov*
*Laboratory of Biological Modeling, NIDDK*
*National Institutes of Health*
*Bethesda, MD, USA*

**Reviewed on OpenReview:** *https://openreview.net/forum?id=dKzReyfUeW*

## Abstract

Local feature attribution is a standard tool for auditing and debugging deep learning predictions, but existing attribution methods are not designed for systems that chain pretrained, frozen, or API-only modules. Gradient-based methods such as Integrated Gradients require an end-to-end computational graph that may be unavailable. Perturbation-based methods such as KernelSHAP require a reference input or background distribution whose choice can substantially alter attributions and may not be defensible for composite pipelines. We present SensX, a local attribution method that treats the model as a black box and replaces arbitrary design choices with interpretable, application-grounded parameters. SensX adapts Morris-style coordinate walks from global sensitivity analysis to local attribution. It requires no access to model internals, training data, or arbitrary reference inputs. We validate SensX across four case studies, each targeting a distinct limitation of existing methods. On a synthetic benchmark where ground-truth relevant features vary per input, SensX reaches 95% top-2 attribution accuracy versus 58% for the best KernelSHAP/Integrated Gradients variant. On a ViT with $> 150,000$ pixel-channel features, SensX produces spatially coherent maps and exposes systematic intra-patch bias where KernelSHAP is infeasible and Integrated Gradients yields task-irrelevant attributions. On single-cell classifiers with unstructured gene-expression features, SensX attains the lowest top-$k$ perturbation AUC. On a composite spatial transcriptomics system where neither method is applicable, SensX reveals reliance on preprocessing grid artifacts and a bias toward low-staining regions.

## 1 Introduction

Local feature attribution aims to identify which input features influence a predictive system's output on a given input, particularly for modern deep learning models. Explanations are typically presented as feature-importance scores, e.g., pixel-level heatmaps in vision and ranked feature lists in tabular settings. Such high-resolution explanations are a practical tool for debugging models, auditing shortcut learning, and assessing whether a system's behavior aligns with domain knowledge. This is especially important in clinical and biomedical settings where model predictions inform diagnoses, treatment decisions, and biological

hypotheses. For example, DeGrave et al. (2021) showed that a high-performing COVID-19 detector relied on dataset-specific artifacts rather than clinically relevant features.

In many AI deployments today, the predictor to be explained is not a single end-to-end network but a composite system built from heterogeneous pretrained, frozen, or API-only components (Wang et al., 2025; Li et al., 2025; Hoang et al., 2024; Nonchev et al., 2025). This breaks two common assumptions behind existing local explainers: (i) access to an end-to-end computational graph for gradients, and (ii) the availability of a defensible reference input or background distribution for perturbation-based methods. While workarounds exist for individual limitations (e.g., differentiable surrogate modeling for gradient access, dataset-based baselines for reference selection, or Shapley-value propagation through model chains), each introduces additional assumptions (access to intermediate representations, distributional choices at module interfaces, or surrogate fidelity) that may not hold in practice.

In particular, gradient-based methods such as Integrated Gradients (Sundararajan et al., 2017) require access to the computational graph, which may be unavailable when individual modules are frozen or served behind an API. Even when gradients are available, they can be numerically unstable in some deep networks (Balduzzi et al., 2017) and flow through frozen encoders whose representations were shaped by pretraining objectives unrelated to the downstream task. The resulting attributions further depend on the choice of integration reference input (Sturmfels et al., 2020), accumulate at irrelevant locations along the integration path (Kapishnikov et al., 2021; Miglani et al., 2020), and can reflect input structure rather than learned model parameters (Adebayo et al., 2018). KernelSHAP (Lundberg and Lee, 2017) and LIME (Ribeiro et al., 2016) are model-agnostic but require a perturbation distribution or background dataset whose specification can substantially affect the resulting attributions (Chen et al., 2022a; Merrick and Taly, 2020), and can be computationally prohibitive (e.g., KernelSHAP triggered a memory allocation request of $\sim 33$ TB for the vision transformer analyzed here). For reference-input- or background-based explainers, the same arbitrariness that affects attribution computation also affects evaluation: validating a ranking requires specifying how to replace feature values, and common choices reintroduce the same reference assumptions.

Global sensitivity analysis offers an alternative, quantifying how input variations affect model output without requiring model internals (Van Stein et al., 2022). However, existing global sensitivity methods rely on arbitrary perturbation regimes. For example, Fel et al. (2021) applied Sobol indices (Sobol, 2001) to neural network explainability by decomposing output variance over perturbation masks applied to image regions. The perturbation masks interpolate each region toward a chosen reference value (e.g., black, gray, or blurred), reintroducing the dependence on an arbitrary reference that limits IG and KernelSHAP. The approach uses a fixed perturbation scale and requires grouping features into spatial regions to remain tractable, making it inapplicable to domains where features lack spatial or hierarchical structure, as in single-cell transcriptomics, where each gene must be attributed individually.

We introduce SensX, a local attribution framework that requires only forward evaluations and admissible feature bounds. SensX identifies an input-specific perturbation scale at which the model response saturates, yielding a neighborhood calibrated to the model's sensitivity rather than chosen ad hoc. Within this neighborhood, it adapts the coordinate-walk strategy of Morris (1991) to local attribution by anchoring trajectories at the input being explained. SensX does not require reference inputs, background datasets, or access to the training distribution. Its hyperparameters are interpretable and grounded in the application.

We validate and demonstrate generality of SensX using four settings, each targeting a distinct limitation of existing methods. The settings span synthetic benchmarks with input-dependent ground truth, a vision transformer with over 150,000 features, single-cell transcriptomics with $\sim 27,000$ genes per cell, and a composite spatial transcriptomics pipeline where gradients are not exposed and background datasets are not available.

## 2 Related Work

**Shapley-based attribution methods.** The Shapley value (Shapley, 1953) provides an axiomatic foundation for additive feature attribution, with model-specific estimators for trees (Lundberg et al., 2020) and deep networks (Lundberg and Lee, 2017), and model-agnostic estimators such as KernelSHAP (Lundberg and

Lee, 2017) and permutation sampling (Štrumbelj and Kononenko, 2010); Covert and Lee (2021) introduced antithetic (paired) sampling for KernelSHAP, reducing estimator variance at fixed evaluation budget, and this is now the default estimator in `shap` $\geq 0.40$. These advances improve estimation efficiency but do not change the feature removal definition and therefore do not affect which features receive attribution mass. Lundberg and Lee (2017) showed that LIME (Ribeiro et al., 2016), DeepLIFT (Shrikumar et al., 2017), and LRP (Bach et al., 2015) are all special cases of the SHAP framework under specific kernel and reference choices, unifying a broad class of attribution methods under a single estimand. Regardless of the estimator, every Shapley-based method must commit to a feature removal definition, marginal expectation, conditional expectation, or baseline substitution, and this choice materially affects the resulting attribution (Chen et al., 2022a; Sturmfels et al., 2020). Conditional SHAP (Aas et al., 2021) addresses reference arbitrariness by conditioning feature removal on the data distribution. This answers a different question from independent perturbation. It identifies which features are important given the co-occurrence structure of the training data, rather than which features the model depends on in isolation. In settings where these two questions diverge, such as composite systems, domain-shifted inputs, or models that have learned distribution-agnostic representations, conditional SHAP reflects data manifold structure while independent perturbation reflects model sensitivity. The Shapley-Taylor interaction index (Dhamdhere et al., 2020) extends this to higher-order feature interactions but is exponential in interaction order and still requires a feature removal definition. For composite pipelines, G-DeepSHAP (Chen et al., 2022b) propagates Shapley values across module interfaces via a rescale rule, but requires model-specific estimators at each module and a shared background distribution at each interface. Frozen encoders and API-only components cannot be handled. Surrogate approaches such as ILLUME (Piaggesi et al., 2025) train interpretable proxies in a learned latent space, but the explanations reflect the surrogate rather than the original system.

**Gradient-based methods.** Integrated Gradients (Sundararajan et al., 2017) accumulates gradients along a straight path from a reference input to $\mathbf{x}$, requiring access to the computational graph and a choice of reference whose effect on the attribution is well-documented (Sturmfels et al., 2020). Grad-CAM (Selvaraju et al., 2017) uses gradients of the output with respect to intermediate spatial feature maps. It is architecture-specific, produces attributions at feature-map resolution rather than individual input features, and is incompatible with forward-only composite systems that do not expose intermediate feature maps.

**Perturbation and masking methods.** RISE (Petsiuk et al., 2018) estimates importance by probing the black-box with randomly masked inputs and constructing a saliency map as a weighted average of those masks. It is model-agnostic and requires no internals, but implicitly uses zero as the reference value for absent features and was designed for spatially structured inputs, making it inapplicable to unstructured feature spaces such as gene expression and not included in the benchmark. Permutation feature importance (Breiman, 2001) measures the average drop in model performance when a feature is randomly permuted across the test set. It is a global method and does not produce per-instance attributions. Counterfactual explanations (Wachter et al., 2017) address a different objective, identifying minimal input changes to alter a prediction, rather than assigning feature importance scores to a given input.

**Learning-to-explain methods.** L2X (Chen et al., 2018) learns to select the $k$ features that maximize mutual information between the selected subset and the model output, training a separate instance-wise selector network on labelled data. INVASE (Yoon et al., 2019) follows a similar paradigm. Both require: (i) $k$ to be specified before attribution, (ii) access to labelled training data and a separate training procedure, and (iii) a globally trained selector that may not reflect the model's local behavior at a specific test input.

**Attention-based methods.** Raw attention maps and attention rollout (Abnar and Zuidema, 2020) are transformer-specific, attribute at token rather than input-feature resolution, and are not output-specific. The same map is produced regardless of which output is being explained. Chefer et al. (2021) addresses the last point by propagating relevance scores through attention layers and skip connections via Deep Taylor Decomposition and LRP, incorporating target-class gradients to produce output-specific maps, but the method still requires full model internals and remains limited to transformer architectures and token resolution.

Table 1: Comparison of feature attribution methods by their core requirements and capabilities.

| Method | Background/ reference | Model- agnostic | Per-feature resolution |
|---|---|---|---|
| Integrated Gradients | Yes | No | Yes |
| DeepSHAP / DeepLIFT | Yes | No | Yes |
| G-DeepSHAP | Per-module | No[*] | Yes |
| KernelSHAP | Yes | Yes | Yes |
| LIME | Yes | Yes | Yes |
| RISE | Yes[#] | Yes | Yes |
| Grad-CAM | No | No | No[†] |
| Attention rollout | No | No | No[¶] |
| Chefer et al. (2021) | No | No | No[¶] |
| L2X / INVASE | No[‖] | Yes | Yes[⋆] |
| Sobol indices | Yes[‡] | Yes | Yes[§] |
| **SensX** | **No[◇]** | **Yes** | **Yes** |

[*]Each module must be a supported type (linear, tree, or standard deep network) with exposed
  internals; cannot be applied to frozen encoders, API-only components, or unsupported architectures.

[#]Implicitly uses zero as the reference value for masked features; designed for spatially
  structured inputs and does not extend naturally to unstructured feature spaces.

[†]Attributes at spatial-feature-map resolution, not individual input features.

[¶]Transformer-specific; attributes at token resolution, not individual input features.

[‡]Perturbation masks interpolate each feature group toward a chosen reference value.

[§]Per-feature in principle, but computationally intractable at high feature counts; existing
  applications (Fel et al., 2021) group features into spatial regions to remain feasible.

[‖]Requires specifying $k$ (number of selected features) before attribution, an implicit design choice analogous to a reference.

[⋆]Produces a subset of $k$ features rather than a full feature ranking per instance.

[◇]Requires a global domain $\Omega^g$ (admissible feature bounds) grounded in domain knowledge. Unlike a background distribution, $\Omega^g$
  does not embed the training distribution and can be specified without access to training data.

**Comparison of requirements.** The common thread across gradient-based, Shapley-based, and masking methods is the need for a reference input or background distribution whose choice is not covered by the method's theoretical guarantees and materially affects the result. SensX requires neither a background dataset nor a reference input. It does require a global domain $\Omega^g$, admissible feature bounds that play a structurally similar role to a reference but differ in kind. The bounds are grounded in domain knowledge of what constitutes a plausible feature value, are auditable, and have predictable consequences for attribution (discussed further in the Discussion). Unlike a background distribution, $\Omega^g$ does not embed the training distribution's structure and can be specified without access to training data. L2X and INVASE also require no reference distribution and are model-agnostic, but require $k$ to be fixed before attribution and a separate training procedure on labelled data.

## 3 SensX

SensX treats the model as a black box that maps input $\mathbf{x} = (x_1, \ldots, x_n)$ to output $\mathbf{y} = m(\mathbf{x})$. The quantity of interest (QOI) is any continuous scalar function $q(\mathbf{y})$. SensX measures the sensitivity of $q(\mathbf{y})$ to each input feature $x_j$ (Figure 1A). For notational convenience, we write $q(\mathbf{x}) \equiv q(m(\mathbf{x}))$.

### 3.1 Local neighborhood and characteristic perturbation scale

Let $\Omega^g = \prod_j [f_j^-, f_j^+]$ be a global domain bounding the input space where $f_j^-$ and $f_j^+$ are lower and upper bounds of the $j$-th input feature. SensX defines a local neighborhood of $\mathbf{x}$ parameterized by a perturbation factor $\delta \in [0, 1]$ as

$$\Omega(\mathbf{x}, \delta) = \prod_{1 \leq j \leq n} \left[ \max\{f_j^-, \, x_j - \delta \, r_j\}, \, \min\{f_j^+, \, x_j + \delta \, r_j\} \right], \quad r_j = f_j^+ - f_j^-. \tag{1}$$

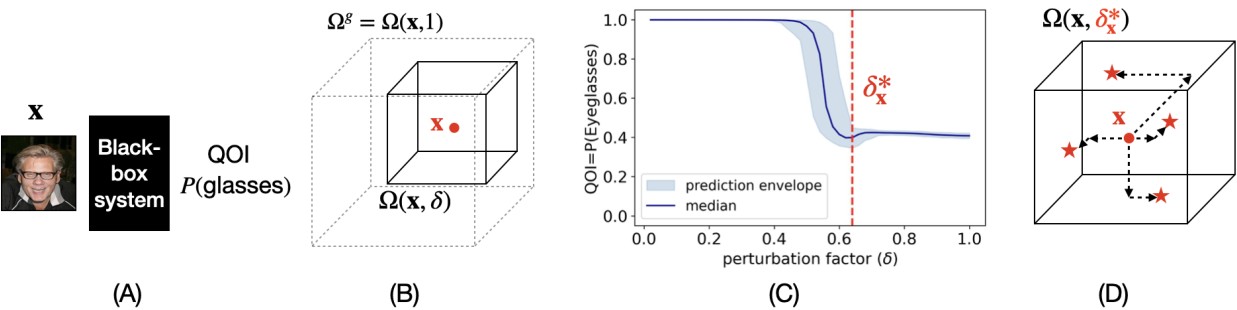

Figure 1: **Overview of SensX.** (A) An input image $\mathbf{x}$ is passed through a black-box system that predicts the probability of wearing eyeglasses (the quantity of interest, QOI). (B) $\Omega(\mathbf{x}, \delta)$ is a local neighborhood parameterized by perturbation factor $0 < \delta \leq 1$ within the global domain $\Omega^g = \Omega(\mathbf{x}, 1)$. (C) The prediction envelope is the variation in QOI (shaded blue). The characteristic perturbation scale $\delta_{\mathbf{x}}^*$ (dashed vertical line) is the smallest $\delta$ beyond which the prediction envelope remains within a user-defined threshold for all larger scales. (D) Within $\Omega(\mathbf{x}, \delta_{\mathbf{x}}^*)$, SensX samples coordinate-wise walks (dashed arrows) from $\mathbf{x}$ to random endpoints (stars) in a random feature order. Changes in QOI along these walks contribute to feature attribution. Algorithm 1 summarizes the main steps of the full framework.

At $\delta = 0$ this reduces to $\{\mathbf{x}\}$; at $\delta = 1$ it equals $\Omega^g$ (Figure 1B). For each $\delta$ in a user-defined discrete set $\Delta$, we draw $n_s$ inputs uniformly at random from $\Omega(\mathbf{x}, \delta)$ and form the distribution of QOIs, $D(\mathbf{x}, \delta, n_s)$. Let $\underline{D}(\mathbf{x}, \delta, n_s)$ and $\overline{D}(\mathbf{x}, \delta, n_s)$ denote the 1st and 99th percentiles of this distribution, which define the prediction envelope (Figure 1C). These percentiles provide a robust measure of the prediction range that mitigates the influence of extreme outliers.

The characteristic perturbation scale $\delta_{\mathbf{x}}^*$ is the smallest $\delta' \in \Delta$ such that the prediction envelope remains within a user-defined threshold $\tau_a$ for all subsequent scales,

$$\delta_{\mathbf{x}}^* = \min_{\delta' \in \Delta} \left\{ \max_{\delta \in \Delta : \delta > \delta'} \overline{D}(\mathbf{x}, \delta, n_s) \; - \; \min_{\delta \in \Delta : \delta > \delta'} \underline{D}(\mathbf{x}, \delta, n_s) \; < \; \tau_a \right\}. \tag{2}$$

This condition is stricter than requiring the envelope width to be small at each individual $\delta$. It ensures the prediction range has globally saturated beyond $\delta_{\mathbf{x}}^*$, not merely at isolated scales (Appendix A.3). The finite-sample stability of the $\delta_{\mathbf{x}}^*$ estimate is analyzed in Appendix A.10. The population saturation criterion is shown to be monotonically non-increasing (Proposition 4), guaranteeing that the calibration identifies a unique transition point. A density-corrected application of the Dvoretzky–Kiefer–Wolfowitz inequality yields an explicit concentration bound for the estimated criterion at fixed grid points, with $n_s$ scaling as $O(\gamma^{-2} f_{\min}^{-2} \log |\Delta|)$ where $\gamma$ is the margin from the threshold and $f_{\min}$ is the minimum QOI density at the relevant percentiles. Discretization error from finite grid spacing is bounded by the local grid spacing near the transition under a strict monotonicity condition (Proposition 5). Appendix B demonstrates this concretely for the single-cell classifiers, where the sensitivity transition occurs near zero: a uniform grid returns $\delta^* = 0$ because all grid points fall in the saturated plateau, while a log-spaced grid concentrates resolution near the transition and recovers a stable non-zero $\delta^*$ across all grid resolutions and sample sizes.

## 3.2 SensX Values

Within the local neighborhood $\Omega(\mathbf{x}, \delta_{\mathbf{x}}^*)$, SensX samples $n_w$ coordinate-wise walks from $\mathbf{x}$ to random endpoints $\mathbf{e}$ uniformly drawn from $\Omega(\mathbf{x}, \delta_{\mathbf{x}}^*)$ (Figure 1D). Each walk is a *grounded trajectory*: a sequence of $n + 1$ points $T_{\mathbf{x},\mathbf{e}} = (\mathbf{t}_0 = \mathbf{x}, \mathbf{t}_1, \ldots, \mathbf{t}_n = \mathbf{e})$ where a random permutation $(i_1, \ldots, i_n)$ determines the order in which features are changed, and $\mathbf{t}_k$ differs from $\mathbf{t}_{k-1}$ only in feature $i_k$, which is set to $e_{i_k}$. This design adapts the Morris method (Morris, 1991) to local attribution by anchoring all trajectories at the input being explained rather than at random points in the input space.

The elementary effect of feature $i_k$ along trajectory $T_{\mathbf{x},\mathbf{e}}$ is the normalized output change at step $k$:

$$d_{i_k}^{T_{\mathbf{x},\mathbf{e}}} = \frac{q(\mathbf{t}_k) - q(\mathbf{t}_{k-1})}{e_{i_k} - x_{i_k}}. \tag{3}$$

The feature order is randomized across trajectories, so each elementary effect is measured against a different configuration of the remaining features.

Following Campolongo et al. (2007), we use the mean absolute elementary effect to avoid sign cancellation when averaging effects across trajectories. The SensX value of feature $j$, computed on grounded trajectories anchored at $\mathbf{x}$ and confined to $\Omega(\mathbf{x}, \delta_{\mathbf{x}}^*)$, is

$$\widehat{s}_{\mathbf{x},j} = \frac{1}{n_w} \sum_{T_{\mathbf{x},\mathbf{e}} \in \mathcal{T}_{\mathbf{x}}} \left| d_j^{T_{\mathbf{x},\mathbf{e}}} \right|. \tag{4}$$

**Grouped attribution.** SensX also supports grouped attribution, where features are partitioned into $G$ disjoint groups and all features in a group are perturbed simultaneously in one trajectory step. This reduces trajectory length from $n$ steps (per-feature) to $G$ steps (per-group) while still sampling independent perturbation targets for each feature within the group. Grouped elementary effects are normalized by the $L_2$ magnitude of the group perturbation to keep sensitivities comparable across groups of equal size (Appendix A.6). Grouped attribution is particularly useful when features have a natural hierarchical or spatial structure, allowing attribution to be resolved at the group level at substantially lower computational cost than full per-feature attribution.

### 3.3 Theoretical Characterization of SensX Values

SensX has two qualitatively distinct perturbation regimes. Formal statements and limitations of both are in Appendix A.11.

$\delta^* \to 0$ **limit.** For a *linear model* $q(\mathbf{x}) = \mathbf{w}^\top \mathbf{x} + b$, the elementary effect of feature $j$ equals $w_j$ along every trajectory for any $\delta^*$ and any $n_w \geq 1$, so $\widehat{s}_{\mathbf{x},j} = |w_j|$ exactly (Proposition 1). For a *nonlinear additive model* $q(\mathbf{x}) = \sum_j g_j(x_j)$, the elementary effect of feature $j$ depends only on the endpoint $e_j$ and not on permutation order or the values of other features. As $n_w \to \infty$ the SensX value converges almost surely to $\mathbb{E}_{e_j}[|g_j(e_j) - g_j(x_j)|/|e_j - x_j|]$, and further to $|g_j'(x_j)|$ as $\delta^* \to 0$ (Proposition 2). For any *continuously differentiable model*, as $\delta^* \to 0$ all endpoints collapse to $\mathbf{x}$, making the permutation order irrelevant and causing the evaluation point for each elementary effect to converge to $\mathbf{x}$ regardless of which features have already been changed. Consequently $\widehat{s}_{\mathbf{x},j} \to |\partial_j q(\mathbf{x})|$ for any fixed $n_w \geq 1$ (Proposition 3, with proofs in Appendix A.8).

$\delta^*$ **bounded away from zero.** SensX is calibrated to operate at the smallest $\delta^*$ at which the model exhibits meaningful sensitivity. For nonlinear models with interactions this value is typically bounded away from zero. The proved results for linear and additive models (Propositions 1 and 2) still hold at any $\delta^*$, but for general non-additive models no analytic form for the estimand exists in this regime. Morris (1991) proposed $\mu_j(\mathbf{x})$, the signed mean of the elementary effects of feature $j$ across trajectories, as a measure of overall influence on the output, and $\sigma_j(\mathbf{x})$, their standard deviation, as a measure of nonlinearity and interactions, since the elementary effect of feature $j$ varies across trajectories when other features modulate its impact. Campolongo et al. (2007) introduced $\mu_j^*(\mathbf{x})$, the mean of the absolute elementary effects of feature $j$, to avoid sign cancellation in $\mu_j(\mathbf{x})$ when the model is non-monotonic. They demonstrated empirically, on the Morris (1991) test function and a $g$-function benchmark, that $\mu_j^*$ is approximately proportional to $S_{Ti}$ (Sobol, 2001), the total-order variance-based sensitivity index inclusive of all interactions. This proportionality is an empirical observation on specific test functions with domain-wide trajectories. $\mu_j^*$ and $S_{Ti}$ measure different mathematical quantities and no theorem establishing a formal relationship between them exists. SensX adopts $\mu_j^*$ and anchors all trajectories at $\mathbf{x}$, the adaptation that turns global sensitivity screening into local feature attribution with respect to a specific input, with $\mathbf{x}$ as the reference point against which all elementary effects are computed. The analytical characterization of the anchored $\mu_j^*$ estimand at non-vanishing $\delta^*$ for general non-additive models remains an open problem (Appendix A.9).

Table 2: Computational cost of SensX across case studies on a single NVIDIA A100 GPU. $n_w$ values are those used in the reported results. The minimum $n_w$ required for stable rankings is determined by the convergence analysis (Figures E1 and E2). Total cost scales linearly with $n_w$.

| Case study | Features $n$ | Walks $n_w$ | Time per walk | Total per input |
|---|---|---|---|---|
| Synthetic (10-d) | 10 | 500–10k | $<1\,\mathrm{s}$ | $<1\,\mathrm{min}$ |
| ViT ($224{\times}224{\times}3$) | 150,528 | 600 | $\sim 5\,\mathrm{min}$ | $\sim 50\,\mathrm{h}$ |
| Single-cell ($\sim$27k genes) | 27,399 | 600 | $0.05\,\mathrm{s}$ | $\sim 30\,\mathrm{s}$ |
| DeepSpot (pixel-level) | 10,000 | 300 | $\sim 1\,\mathrm{min}$ | $\sim 5\,\mathrm{h}$ |

## 3.4 Implementation details

**Hyperparameters.** Our implementation evaluates multiple feature updates per forward pass by batching features. Batch size trades off GPU throughput and memory.

The global domain $\Omega^g$ is specified by the practitioner based on domain knowledge of admissible feature ranges. The parameters $n_s$ and $n_w$ control convergence of the stability profile and attribution estimate, respectively, and the discretization of the $\delta$ sweep controls the resolution of the stability profile. The primary context-dependent choice is the stability threshold $\tau_a$, which specifies what magnitude of output change is considered meaningful. For classification tasks, $\tau_a = 0.1$ corresponds to a 10% change in predicted probability. For gene expression prediction, $\tau_a = 0.5$ corresponds to a half-unit change in log expression. These choices are transparent, auditable, and grounded in the application. Algorithm 1 summarizes the main steps of the full framework.

**Computational cost.** Each grounded trajectory requires evaluating the QOI at $n$ points, one per feature update. With batch size $B$, these $n$ evaluations are grouped into $\lceil n/B \rceil$ forward passes, since the endpoint $\mathbf{e}$ is reached at step $n$ and is therefore included in the final batch. Across $n_w$ independent walks, the total number of forward evaluations is $n_w \cdot \lceil n/B \rceil$, and the total wall-clock time is $\mathcal{C} \approx n_w \cdot \lceil n/B \rceil \cdot T$, where $T$ is the time to complete one forward pass over a batch of $B$ feature updates. Increasing $B$ reduces $\lceil n/B \rceil$ and therefore $\mathcal{C}$, at the cost of higher peak memory. The batch size thus controls the throughput–memory tradeoff. The walks are fully independent and trivially parallelizable across devices, reducing wall-clock time proportionally to the number of devices.

# 4 Experiments

## 4.1 SensX recovers local feature attributions that reference-based methods miss

We benchmark SensX against KernelSHAP and Integrated Gradients (IG) with three reference-input choices (zero vector, training-set mean, and random training samples) on four established synthetic datasets with known relevant features (Chen et al., 2018) (Appendix F.1). KernelSHAP is feasible at the synthetic feature count, so we use it as the model-agnostic Shapley baseline. DeepSHAP is omitted as it offers no advantage over KernelSHAP when the latter is tractable.

Figure 2 shows results on the Switch dataset, where relevant features vary dynamically per input. SensX outperforms all alternatives at every $k$. At top-2, SensX (95%) exceeds the best alternative, IG (random), by 37 percentage points. At top-5, accuracy drops below 13% for all methods, reflecting the difficulty of identifying all five relevant features simultaneously. Results are robust to hyperparameter choice, with accuracy varying by fewer than 5 percentage points across the full hyperparameter range for all methods (Figures D1 and D2). On the remaining datasets, both SensX and KernelSHAP achieve perfect accuracy on XOR (Figure D1A). On Orange Skin and Nonlinear Additive, the methods diverge only at the highest $k$, where SensX converges to near-perfect accuracy while KernelSHAP plateaus below ( Figures D1B and D1C).

---

**Algorithm 1** SensX($\mathbf{x}$)

---

**Require:** Scalar QOI $q(\cdot)$, input $\mathbf{x} \in \mathbb{R}^n$, global bounds $\Omega^g = \prod_j [f_j^-, f_j^+]$, sweep set $\Delta$, samples $n_s$, walks $n_w$, stability threshold $\tau_a$
**Ensure:** SensX values $\widehat{\mathbf{s}}_{\mathbf{x}} \in \mathbb{R}^n$
1: Define local neighborhoods $\Omega(\mathbf{x}, \delta)$ $\hfill \triangleright$ equation 1
2: **Stability profile / envelope:**
3: **for all** $\delta \in \Delta$ **do**
4:     Sample $n_s$ perturbations $\sim \text{Unif}(\Omega(\mathbf{x}, \delta))$
5:     Evaluate QOI values and record the 1st/99th percentiles
6: **end for**
7: **Select characteristic scale:**
8: Compute $\delta_{\mathbf{x}}^*$ from the envelope and handle edge cases $\hfill \triangleright$ equation 2
9: **Grounded coordinate-walk attribution (anchored $\mu^*$):**
10: Initialize $S_j \leftarrow 0$ for all $j = 1, \ldots, n$
11: **for** $w = 1, \ldots, n_w$ **do**
12:     Sample endpoint $\mathbf{e} \sim \text{Unif}(\Omega(\mathbf{x}, \delta_{\mathbf{x}}^*))$
13:     Sample a random permutation $\pi$ of $\{1, \ldots, n\}$
14:     Set $\mathbf{t} \leftarrow \mathbf{x}$
15:     **for** $k = 1, \ldots, n$ **do** $\hfill \triangleright$ Batched (GPU-parallelism)
16:         $i \leftarrow \pi(k)$
17:         Update $t_i \leftarrow e_i$ (change one feature at a time)
18:         Compute elementary effect for feature $i$ and accumulate magnitude: $\hfill \triangleright$ equation 3
19:             $S_i \leftarrow S_i + |d_i|$
20:     **end for**
21: **end for**
22: **SensX values:**
23: $\widehat{s}_{\mathbf{x},j} \leftarrow S_j / n_w$ for all $j$
24: **return** $\widehat{\mathbf{s}}_{\mathbf{x}}$

---

The characteristic perturbation scale $\delta^*$ converges to 1 for nearly all inputs across three of the four datasets (Figure D3B). On the Orange Skin dataset, $\delta^*$ shows non-trivial variance at low sample budgets before stabilizing near 1 (Figure D3A). In the higher-dimensional case studies that follow, $\delta^*$ takes values below 1.

To verify that the SensX advantage is not attributable to estimator choice, we additionally benchmarked `PermutationExplainer` (Covert and Lee, 2021) at the same evaluation budget as KernelSHAP. SensX outperforms both Shapley estimators across all Top-$k$ values (Figure 2), confirming that the performance gap is driven by the feature removal assumption, not estimator variance.

## 4.2 Feature-level attribution of a vision transformer reveals architectural bias

We finetuned two pretrained ViT models (Dosovitskiy et al., 2020) on the CelebFaces Attributes dataset (Liu et al., 2015) for smile detection and eyeglass detection (Appendix F.2). Input images are $224 \times 224$ pixels across three color channels, totaling 150,528 features, where each feature is an individual color channel at a pixel location. The characteristic perturbation scale ranges from $\delta^* = 0.54$ to 0.70 across the four (image, model) combinations. KernelSHAP required 33 TB of memory at this scale, whereas SensX required less than 10 GB. Gradient access is available for these models, so we also computed IG with zero vector, training-set mean, and expected gradients (Erion et al., 2021) reference inputs as a non-model-agnostic comparison. DeepSHAP was not applied as it relies on backpropagation rules designed for standard feedforward layers and is not straightforwardly applicable to transformer architectures with attention and layer normalization.

Fig. E1 shows how the SensX feature ranking stabilizes with the number of walks. The Spearman correlation between rankings computed with $n_w$ and $n_w - 20$ walks exceeds 0.99 by $n_w \approx 200$ across all (image, model) pairs. Each walk takes $\sim 5$ minutes with batch-size of 1000 on a single NVIDIA A100 GPU at 32-bit precision for the ViT forward evaluation. The following results in this section are based on $n_w = 600$.

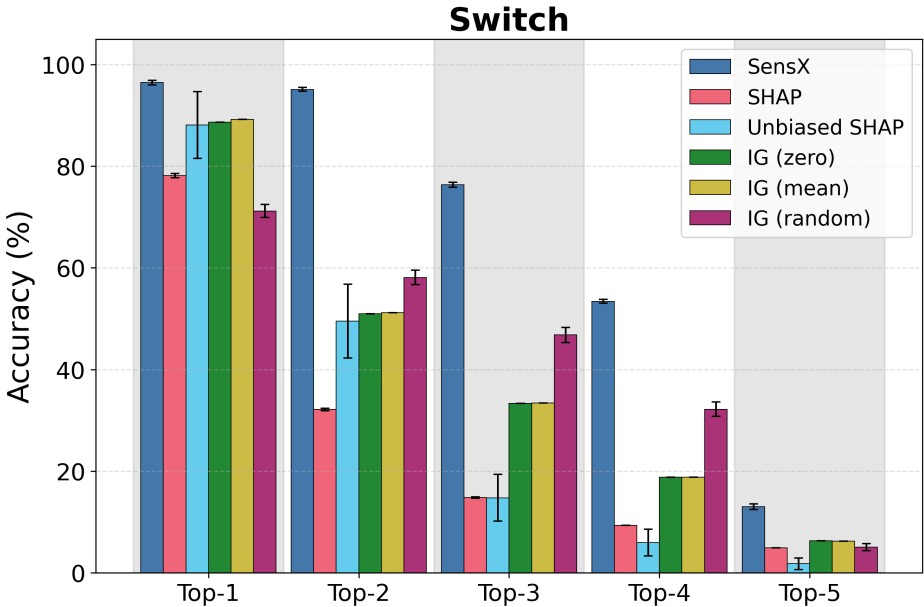

Figure 2: **SensX recovers local feature attributions that reference-dependent methods miss.** Fraction of samples in which the top-$k$ ranked features all belong to the ground-truth set, averaged over 100 independent runs ($\pm 1$ standard deviation), at the best-performing hyperparameter for each method on the Switch dataset. SensX outperforms KernelSHAP, Unbiased SHAP, and all three IG reference-input variants across all $k$ values.

Figure 3A shows masks of the top-$k$ ranked SensX features. For the eyeglasses model, the top 2,500 features localize tightly to the eyeglass frames and for the smiling model, they localize to the mouth. IG attributions are reference-input-dependent and mislocalize to task-irrelevant regions (Figure 3B). On the smiling model, the mean reference input and expected gradients (Erion et al., 2021) attribute to high-contrast eyeglass frames rather than the mouth, consistent with the input-structure dependence identified by Adebayo et al. (2018). All IG reference-input variants lack the spatial coherence of SensX attributions (Figure D5).

We validated the SensX ranking with two complementary checks. First, perturbing features from highest to lowest SensX rank produces a rapid QOI decline, while perturbing over 130,000 bottom-ranked features has no measurable effect (Figure D6A). Second, the cascading randomization test of Adebayo et al. (2018) confirms that attributions depend on learned weights rather than input structure. Progressively randomizing transformer blocks destroys the SensX ranking, and at deeper randomization levels the characteristic perturbation scale $\delta^*$ drops to zero, indicating that the model output becomes insensitive to input perturbations (Figure D6B,C).

The SensX heatmaps reveal grid artifacts aligned with the $16 \times 16$ pixel patches defining the ViT tokenization. Patch-normalized SensX is consistently highest at the patch center, decays toward the interior boundary, and rebounds at the edge (Figure 4). This profile holds across all color channels and all image–model combinations ( Figure D8), indicating a systematic architectural bias rather than image-specific structure. IG attributions analyzed with the same patch-normalization methodology show a coarse center bias in the distance profiles, but the pattern is not apparent in the normalized IG heatmaps (Figure D9), suggesting that the intra-patch structure is below the effective resolution of IG at this scale.

### 4.3 SensX rankings concentrate model sensitivity into fewer genes than DeepSHAP and Integrated Gradients in single-cell classification

We trained 14 binary classifiers, one per cell type, on the core human lung cell atlas (Sikkema et al., 2023) (Appendix F.3). The characteristic perturbation scale $\delta^*$ varies by up to two orders of magnitude across cell

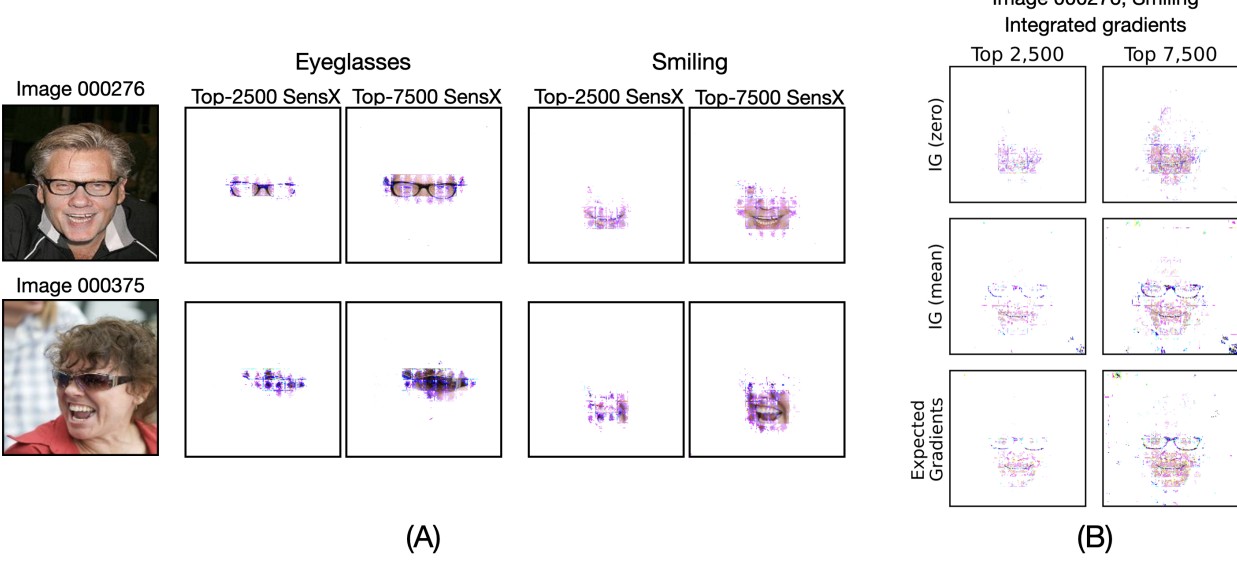

Figure 3: **SensX produces spatially coherent attributions at over 150,000 features.** (A) Top-$k$ SensX masks for two input images under the eyeglasses and smiling models. Top-ranked features concentrate on class-relevant regions and expand coherently as $k$ increases from 2,500 to 7,500. Features outside the top-$k$ are set to white, where each feature is an individual color channel at a pixel location. (B) IG masks for image 000276 under the smiling model with three reference-input choices. Two of the three reference inputs attribute to the eyeglass frames rather than the mouth, despite the smiling model being fine-tuned independently. All masks are shown in Figure D5.

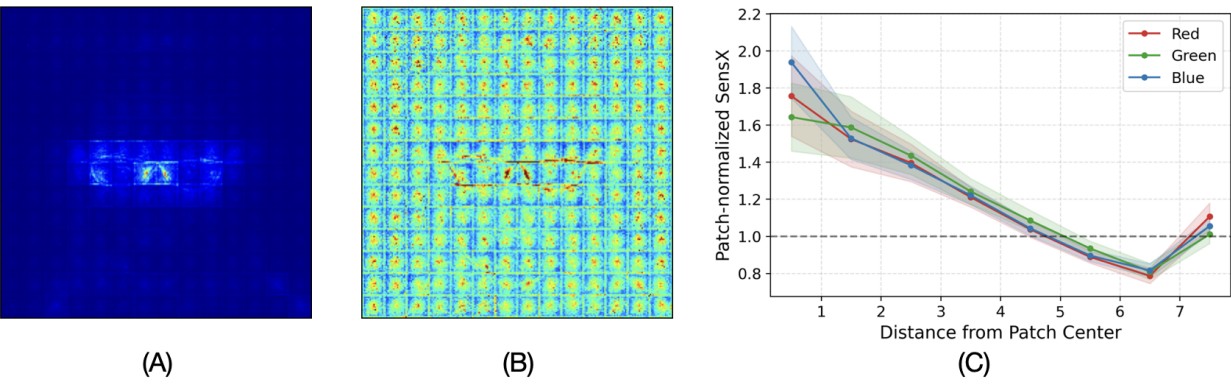

Figure 4: **Precise attribution exposes intra-patch spatial bias in ViT tokenization.** (A) SensX attribution heatmap (green channel, Eyeglasses model, image 000276). (B) The same heatmap after normalizing each $16 \times 16$ patch by its median, revealing a consistent intra-patch pattern. (C) Patch-normalized SensX averaged across all patches as a function of distance from the patch center. Shaded regions denote $\pm 1$ standard deviation.

types (Figure D10). Lower values indicate that the classifier's predictions are determined by finer differences in expression. The within-cell-type spread shows that SensX adapts $\delta^*$ independently for each cell.

The global domain $\Omega^g$ was set to the feature-wise minimum and maximum of the training data. For log-normalized gene expression data, however, biologically grounded bounds are also available independently of any specific dataset: the lower bound is exactly zero for all genes, since a gene that is not expressed has zero

counts and $\log(1+0) = 0$ is a true biological state, and the upper bound is constrained by the normalization scheme itself: for log1p-normalized counts-per-10k, the theoretical maximum is $\log(1 + 10{,}000) \approx 9.2$. A practitioner without access to training data could therefore define $\Omega^g = [0, 10]^n$ for all genes, which is biologically defensible and would yield the same practical result.

For a rank-convergence analysis, we computed SensX rankings cumulatively and measured the Spearman correlation between rankings obtained with $n_w$ and $n_w - 20$ walks (in increments of 20). Mean correlations across 1000 cells per cell type rise quickly toward 1, while the across-cell standard deviation decreases with additional walks, indicating that rankings stabilize both on average and across cells (Figure E2). The following results use $n_w = 600$ walks. The computational cost for single-cell attribution is significantly lower as compared to the ViT. SensX processes 20 coordinate-walks per second for a single cell ($27{,}399$ features) using a batch size of $8{,}192$. This represents a throughput nearly three orders of magnitude higher than the ViT case study.

We compared SensX with DeepSHAP (Lundberg and Lee, 2017) and Integrated Gradients (Sundararajan et al., 2017) with three reference-input variants. KernelSHAP requires over $230\,\mathrm{GB}$ of memory and PermutationExplainer is computationally intractable at this feature count. DeepSHAP is the viable Shapley-based alternative since the classifiers are standard feedforward networks. For each cell, genes are ranked by the absolute value of their attribution. All methods attribute the same quantity of interest (predicted probability).

Figure 5A shows perturbation curves for natural killer cells at $\delta^*$. The predicted probability drops fastest with the fraction of top-ranked genes perturbed for SensX, and requires the largest fraction of bottom-ranked genes perturbed before changing. Figure 5B summarizes the normalized AUC across all cell types. SensX achieves the lowest top-$k$ AUC (0.53–0.83) and the highest bottom-$k$ AUC (0.98–1.00) across all cell types.

The perturbation validation at $\delta^*$ uses a scale derived from SensX's own stability analysis. We repeated the analysis at $\delta = 1$ using the same SensX rankings ( Figure D11). SensX top-$k$ AUC is zero (rounded to two decimal places) for all cell types, showing that the rankings transfer to global perturbation.

The comparison across IG reference-input variants shows that reference-input choice has a substantial effect on attribution quality. IG with a training-data mean reference input outperforms IG with the mean of the high-confidence cells in every cell type (Figure 5B), despite the latter being closer to the inputs being explained, an instance of the reference-input sensitivity discussed in Section 4.2 and examined further in the Discussion.

### 4.4   Pixel-level attribution of a composite spatial transcriptomics model

DeepSpot (Nonchev et al., 2025) is a composite framework that predicts expression of 5,000 genes from H&E histology images. For a given spot ($100 \times 100$ pixel tile), DeepSpot integrates the tile and its eight neighbors through the UNI pathology foundation model (Chen et al., 2024) and secondary architectures (Figure 6A, Appendix F.4). We show results only for SensX. IG is not applicable because UNI is frozen (no exposed computational graph or gradients). KernelSHAP is also not applicable in a principled way because the training sets of the system and its modules are not known. We applied SensX to 745 spots from a whole-slide H&E colon tissue image, analyzing four genes selected for high mean and high variance of predicted log-expression.

We first attributed at the tile level by grouping 270,000 input features into 9 tiles. The center tile's SensX value exceeds that of the strongest neighbor by more than an order of magnitude in 98.4% of spot-gene pairs, confirmed by independent center-tile perturbation validation (Figure D12). We then computed pixel-level SensX values within the center tile. The characteristic perturbation scale ranges from $\delta^* = 0.1$ to $0.6$ across spots and genes ( Figure D13). Perturbation validation confirms that the pixel-level ranking is effective (Figure D15).

The pixel-level attributions reveal two systematic patterns (Figure 6). First, high-attribution pixels concentrate near the boundaries of the $3 \times 3$ subspot grid used in DeepSpot preprocessing. For each spot-gene pair we test the null hypothesis that high-attribution pixels are distributed uniformly over the center tile against the one-sided alternative that they concentrate closer to grid lines than expected under uniform sampling (one-tailed permutation test; null distribution of mean grid-line distances from 500 uniformly sampled pixels,

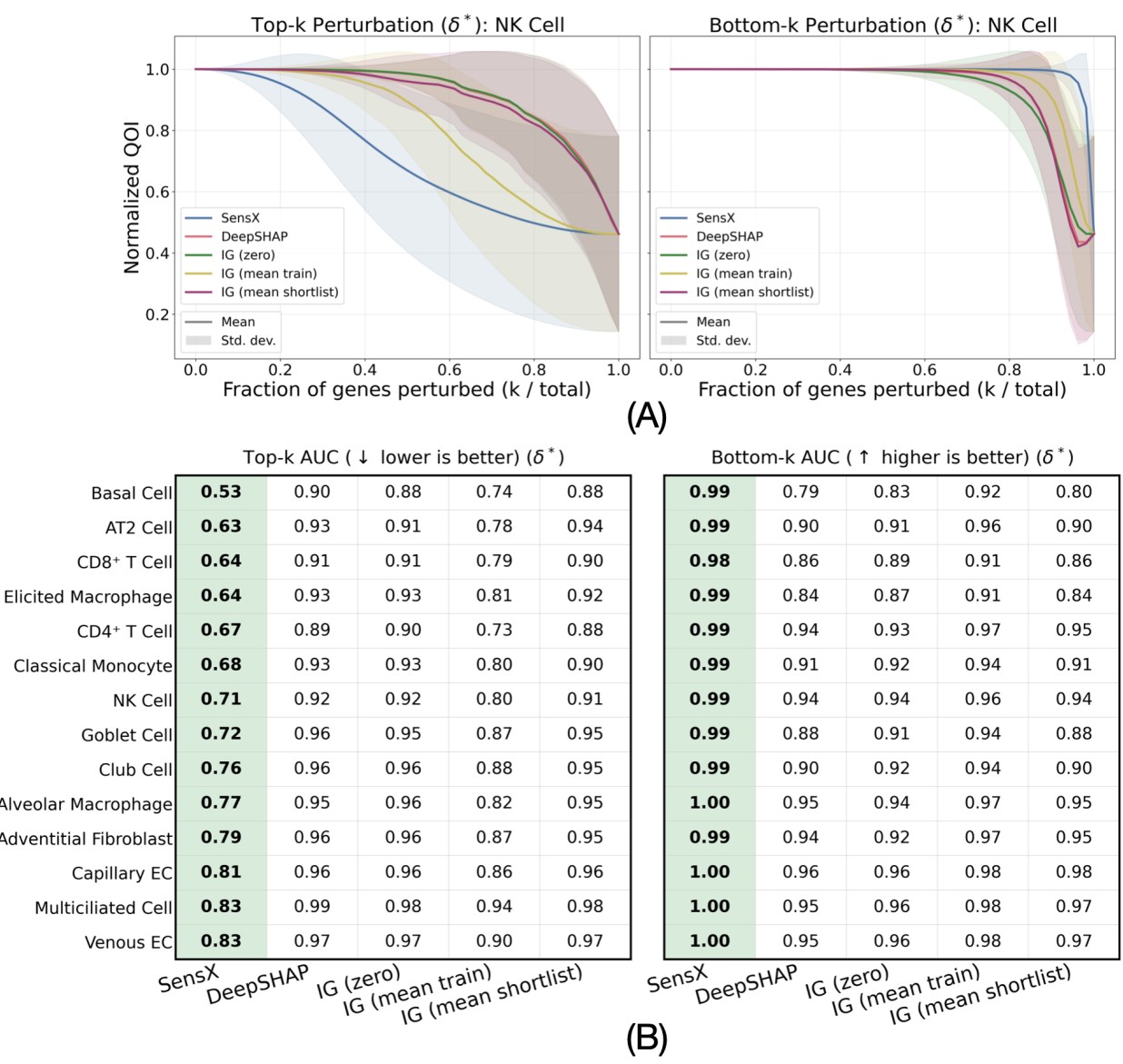

Figure 5: **Perturbation validation of per-cell feature rankings across 14 cell types.** (A) Top-$k$ (left) and bottom-$k$ (right) perturbation curves for natural killer cells. Curves show the mean across 1,000 cells, with shaded regions denoting $\pm 1$ standard deviation. (B) Normalized AUC of the perturbation curves for all 14 cell types. For top-$k$, lower is better. For bottom-$k$, higher is better. SensX achieves the best value (bold) in every cell type for both metrics.

10,000 permutations, computed once and shared across all spot-gene pairs): 61.9% of spot-gene pairs are significant at $p < 0.025$, ranging from 60.8% ($LGALS4$) to 63.6% ($MUC5B$) ( Figure D14A), confirming that the subspot partitioning introduces spatial bias. Second, among spot-gene pairs without significant grid bias, we test the null hypothesis that high-attribution pixels are distributed uniformly over the center tile against the one-sided alternative that they have lower staining saturation than expected under uniform sampling (one-tailed permutation test; null distribution of mean saturation from 500 uniformly sampled pixels, 10,000 permutations per spot, shared across genes): 73.9% are significant at $p < 0.025$, ranging from 71.0% ($LGALS4$) to 80.8% ($TFF3$) (Figure D14B), indicating that the model preferentially attends to lighter-stained regions. These low-saturation regions are visually consistent with colonic crypts (Figure 6B).

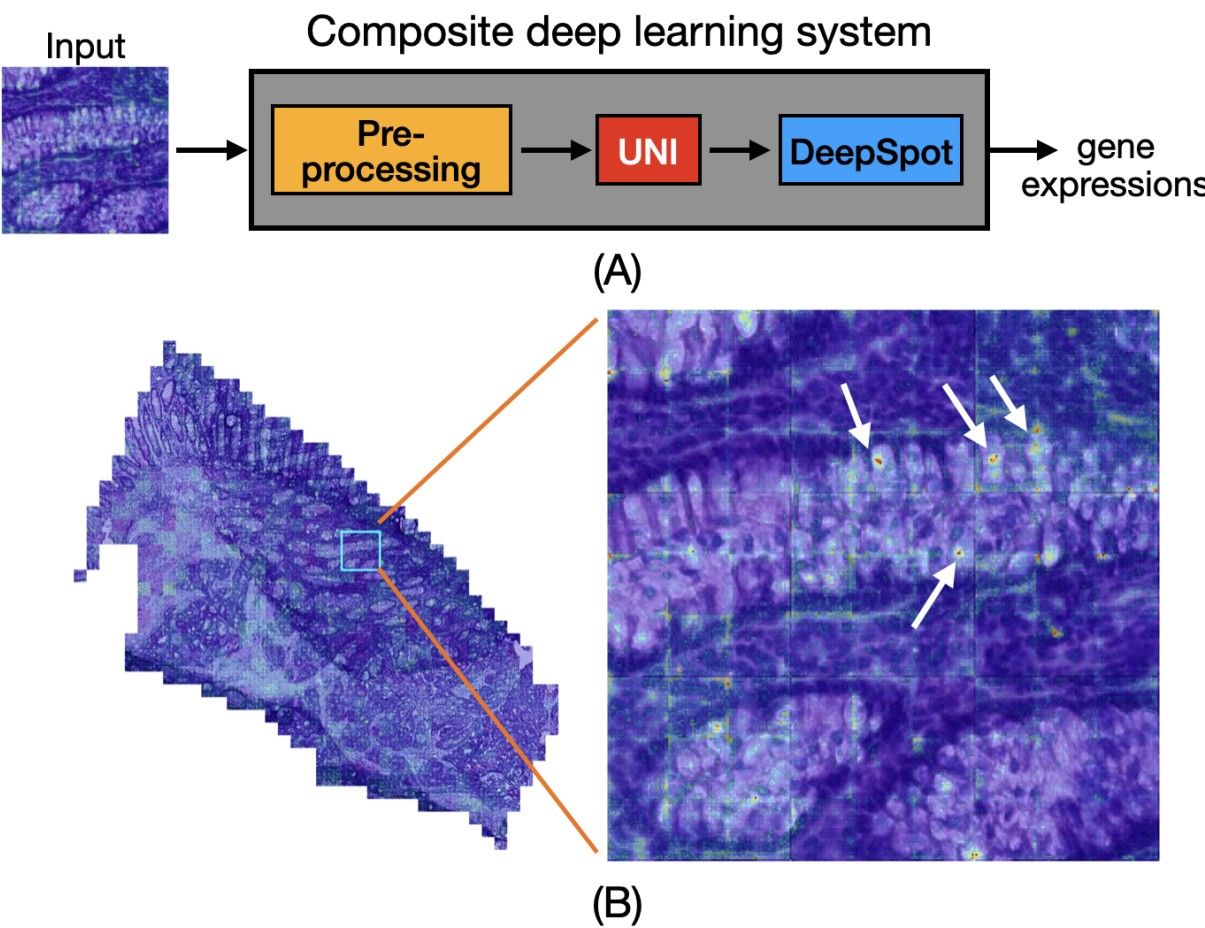

Figure 6: **Pixel-level SensX attributions for gene expression predictions of a composite spatial transcriptomics model.** (A) The composite system chains three modules: preprocessing to get subspots, frozen pretrained foundation model UNI to get morphological features, and custom deep neural network (DeepSpot) to predict gene expression. (B) Whole-slide view SensX attribution overlay for *MUC4* expression in a colon tissue section. The cyan box is zoomed in to show boundary effects that align with the $3 \times 3$ subspot grid that is created during preprocessing. White arrows indicate regions of lighter staining where high-attribution pixels concentrate.

## 5 Discussion

**Attribution granularity.** The case studies collectively show that the granularity of attribution determines what can be discovered about a model's behavior. The ViT patch-boundary bias and the DeepSpot subspot grid effects are invisible to any method that attributes at the token or region level. They become visible only when attribution is resolved to individual pixel-channels and pixels, respectively. In single-cell transcriptomics, where genes lack spatial or hierarchical structure, feature-level attribution is the only option, and finer resolution directly translated into rankings that concentrated model sensitivity into fewer genes than all alternatives tested. These artifacts would go undetected by coarser methods regardless of their theoretical properties.

**Manifold objection.** An objection to independent feature perturbation is that it produces unrealistic feature combinations off the data manifold (Frye et al., 2020). This concern reflects a distinction between two different objectives: explaining predictions under the data distribution and identifying which features the

model depends on. Data-aware methods such as conditional SHAP (Aas et al., 2021) address the former by spreading attribution across correlated features, reflecting the dataset's covariance structure. Independent perturbation addresses the latter by isolating whether the model has learned to rely on one feature over another. SensX operates in this second setting. The $\delta^*$ calibration constrains perturbations to the model's local sensitivity regime rather than conditioning on the data distribution.

**Relationship to attribution axioms.** SensX satisfies several classical attribution desiderata: *sensitivity* (a feature that can alter $q$ within $\Omega(\mathbf{x}, \delta_{\mathbf{x}}^*)$ receives a nonzero value in expectation), *implementation invariance* (two models with identical input-output behavior receive identical attributions, since SensX is black-box), *symmetry* in expectation (two functionally interchangeable features receive equal expected values via permutation-averaging), and the *dummy* property (a feature with no effect on $q$ receives zero in expectation). SensX does not satisfy *completeness* (attributions do not sum to the prediction difference from a reference) and does not provide directional information, both consequences of using mean absolute elementary effects. The intended use is ranking-oriented. The high-$\mu_j^*$ features identified by SensX provide a reduced subset on which sign-sensitive or directional analyses can subsequently be applied at far lower cost than on the full feature set.

**Hyperparameter interpretability.** The three user-specified inputs to SensX, the stability threshold $\tau_a$, the perturbation factor grid $\Delta$, and the global domain $\Omega^g$, are interpretable and grounded in the application, which distinguishes them from the arbitrary choices embedded in competing methods.

**Stability threshold $\tau_a$.** $\tau_a$ encodes a domain-meaningful tolerance for output variation. In the limit of infinite $n_s$ and infinitely fine $\Delta$ for a smooth model, $\delta_{\mathbf{x}}^*$ is non-decreasing as $\tau_a$ decreases. A stricter threshold requires the model to have saturated more tightly before $\delta^*$ is declared. In practice, finite grid resolution and sampling noise mean the effect of changing $\tau_a$ on $\delta_{\mathbf{x}}^*$ depends on the model's sensitivity profile and $\Delta$. When $\tau_a$ is too strict, $\delta_{\mathbf{x}}^*$ may equal zero for inputs in flat prediction regions. As the $\delta^*$ diagnostic paragraph below shows, this is an explicit flag rather than a silent failure. When $\tau_a$ is too loose, $\delta^* \to 1$ and perturbations span the full domain $\Omega^g$. A detailed empirical characterization of the joint dependence of $\delta^*$ on $\tau_a$, $k$, and $n_s$ for the single-cell classifier, including single-cell and population-level convergence analysis across all 14 cell types, is provided in Appendix B.

**Perturbation factor grid $\Delta$.** The choice of $\Delta$ determines the resolution at which the stability profile is estimated. An initial uniform grid sufficed in most settings. We used a log-spaced grid for the single-cell case since $\delta^*$ clustered near zero. The resolution can always be refined in the neighborhood of the first estimate of $\delta^*$ by a secondary stability sweep.

**Global domain $\Omega^g$.** Reference-based methods depend on a background dataset or reference input whose choice has unpredictable effects on attribution and lacks a principled criterion (Figure 3B, Figure 5B). $\Omega^g$ plays a structurally similar role but is interpretable. It encodes admissible feature ranges grounded in domain knowledge, with predictable consequences. Setting it too tightly risks missing genuine model sensitivity, while setting it too broadly includes implausible feature values. The sensitivity of $\delta^*$ and feature rankings to $\tau_a$, $\Delta$, and $\Omega^g$ is inherently model- and input-dependent. The perturbation validation protocol (Appendix A.7) provides a model-specific check that rankings are meaningful for any given parameter setting.

**$\delta^*$ as a diagnostic.** $\delta_{\mathbf{x}}^* = 0$ flags inputs where model output variation is below threshold under any neighborhood size, warranting investigation rather than producing misleading attributions. This is preferable to methods that silently produce attributions regardless of whether the model exhibits meaningful local sensitivity.

**Comparison with gradient-based methods.** The coordinate-wise walks in SensX differ structurally from path-based methods such as IG. IG interpolates all features simultaneously along a straight path from a reference input, accumulating gradients that can vanish in saturated regions. SensX changes one feature at a time in random order and measures finite output differences, averaging interaction effects across multiple walks. This structural difference, together with the $\delta^*$ calibration, is what allows SensX to resolve the

intra-patch bias in the ViT and the subspot grid effects in DeepSpot. Both patterns require perturbations confined to the sensitivity regime and resolved at the level of individual features.

**Limitations.** *Theoretical.* At finite $\delta^*$, no analytic characterization of the SensX estimand exists for general non-additive models. This is the open problem stated in Appendix A.11. In the $\delta^* \to 0$ limit, SensX values converge to local partial derivatives, a first-order quantity that does not capture interaction effects. The $\mu_j^*$ statistic, and by inheritance anchored SensX, tends to underestimate features involved in strong interactions under limited sampling. The perturbation validation protocol (Appendix A.7) provides an empirical check for this case.

Axiomatic. As noted above, SensX does not satisfy completeness and lacks directional information. It is not an axiomatic decomposition of the prediction. The intended use is identifying a reduced feature subset on which sign-sensitive methods can be applied at lower cost.

*Practical.* Computational cost scales linearly with $n_w$. The required $n_w$ is determined by the ranking convergence analysis or perturbation validation protocol rather than a fixed budget. The coordinate-wise walk structure does not extend naturally to discrete inputs such as binary, ordinal, or categorical features. A reformulation is left to future work. Grouped attribution is currently restricted to equal-sized groups, as the $L_2$ normalization of grouped elementary effects is not comparable across groups of different sizes. Extending to unequal group sizes is left to future work.

## Code and data availability

The source code for our implementation, including the training scripts, hyperparameter configurations, data simulation or data acquisition details for our case studies, is provided in the supplementary material. SensX is also publicly available at https://github.com/nihcompmed/SensX.

## Acknowledgments

This research was supported by the Intramural Research Program of the National Institute of Diabetes and Digestive and Kidney Diseases (NIDDK) within the National Institutes of Health (NIH). The contributions of the NIH author(s) were made as part of their official duties as NIH federal employees, are in compliance with agency policy requirements, and are considered Works of the United States Government. However, the findings and conclusions presented in this paper are those of the author(s) and do not necessarily reflect the views of the NIH or the U.S. Department of Health and Human Services. This work utilized the computational resources of the NIH HPC Biowulf cluster (https://hpc.nih.gov).

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

# A  SensX Framework: Formal Definitions

## A.1  Terminology

A trained model is a mathematical function that maps an input to an output. $\mathbf{x} = (x_1, \ldots, x_n)$ is a model input of $n$ features (all real numbers) and $\mathbf{y} = m(\mathbf{x})$ is the corresponding model output. The quantity of interest (QOI) for GSA is a scalar function (a single real number) of the model output, which is a function of the input,

$$q(\mathbf{x}) \equiv q(m(\mathbf{x})) \equiv q(\mathbf{y}). \tag{A1}$$

## A.2  Global Domain and Parameterized Local Neighborhood

We define the global domain as

$$\Omega^g = \prod_{1 \leq j \leq n} [f_j^-, f_j^+] \tag{A2}$$

where $f_j^-$ and $f_j^+$ are lower and upper bounds of the $j$-th feature. The framework accepts any user-defined $\Omega^g$ that may be contextually relevant based on expert advice. For the case studies in this work, $\Omega^g$ was chosen as the smallest hypercube containing the training set.

We define a local neighborhood of $\mathbf{x} \in \Omega^g$ parameterized by perturbation factor $\delta \in [0, 1]$ as

$$\Omega(\mathbf{x}, \delta) = \prod_{1 \leq j \leq n} [\max\{f_j^-, x_j - \delta r_j\}, \min\{f_j^+, x_j + \delta r_j\}], \text{ where } r_j = f_j^+ - f_j^-. \tag{A3}$$

From Equation A3,

$$\Omega(\mathbf{x}, 0) = \{\mathbf{x}\}, \tag{A4}$$
$$\Omega(\mathbf{x}, 1) = \Omega^g, \text{ and} \tag{A5}$$
$$\Omega(\mathbf{x}, \delta_1) \subseteq \Omega(\mathbf{x}, \delta_2) \text{ if } \delta_1 < \delta_2. \tag{A6}$$

## A.3  Identifying Input-Specific Characteristic Perturbation Scale

To avoid arbitrary selection of a perturbation scale, SensX identifies a characteristic perturbation factor $\delta_{\mathbf{x}}^*$ for each input $\mathbf{x}$. Let $D(\mathbf{x}, \delta, n_s)$ be the distribution of QOIs at $n_s$ uniformly random samples within $\Omega(\mathbf{x}, \delta)$. We compute this distribution for all $\delta$ in a user-defined discrete set of perturbation factors $\Delta$. By default, $n_s = 1000$ and $\Delta = \text{linspace}(0.02, 1, \text{num} = 50)$. Let $\underline{D}(\mathbf{x}, \delta, n_s)$ and $\overline{D}(\mathbf{x}, \delta, n_s)$ be the 1st and 99th percentiles of $D(\mathbf{x}, \delta, n_s)$, respectively.

We define the characteristic perturbation factor $\delta_{\mathbf{x}}^*$ as the smallest scale beyond which the prediction envelope remains within a user-defined threshold $\tau_a$ for all subsequent scales:

$$\delta_{\mathbf{x}}^* = \min_{\delta' \in \Delta} \left\{ \max_{\delta > \delta'} \{\overline{D}(\mathbf{x}, \delta, n_s)\} - \min_{\delta > \delta'} \{\underline{D}(\mathbf{x}, \delta, n_s)\} < \tau_a \right\}. \tag{A7}$$

Intuitively, $\delta_{\mathbf{x}}^*$ is the smallest neighborhood size at which the model's response to perturbations reaches a steady state. The region $\Omega_{\mathbf{x}}^* = \Omega(\mathbf{x}, \delta_{\mathbf{x}}^*)$ constitutes the input-specific neighborhood where the model remains sensitive to feature variations.

### A.4 Adapting the Morris Method for Local Attribution Through Grounded Trajectories

In the standard Morris method, trajectories begin at random points in the input space, producing global sensitivity measures. SensX adapts this design for local feature attribution by anchoring all trajectories at the input $\mathbf{x}$ being explained.

For each of $n_w$ perturbation endpoints $\mathbf{e} = (e_1, \ldots, e_n)$ uniformly sampled from $\Omega_{\mathbf{x}}^*$, we define a grounded trajectory as a sequence of points starting at $\mathbf{x}$ and ending at $\mathbf{e}$, where one feature value in $\mathbf{x}$ is replaced by its value in $\mathbf{e}$ at each step. The order in which features are perturbed is randomly selected. Formally, a grounded trajectory $T_{\mathbf{x},\mathbf{e}}$ is a sequence of $n+1$ points

$$T_{\mathbf{x},\mathbf{e}} = (\mathbf{t}_0 = \mathbf{x}, \mathbf{t}_1, \ldots, \mathbf{t}_n = \mathbf{e}), \tag{A8}$$

where $(i_1, \ldots, i_n)$ is a random permutation of feature indices and $\mathbf{t}_k = (t_{k1}, \ldots, t_{kn})$ such that

$$t_{kj} = \begin{cases} e_j & \text{if } j \in \{i_1, i_2, \ldots, i_k\} \\ x_j & \text{otherwise.} \end{cases} \tag{A9}$$

The set of $n_w$ grounded trajectories originating at $\mathbf{x}$ is denoted $\mathcal{T}_{\mathbf{x}}$.

### A.5 SensX Values and SensX Vector

The elementary effect of feature $i_k$ along a grounded trajectory $T_{\mathbf{x},\mathbf{e}} \in \mathcal{T}_{\mathbf{x}}$ is defined as

$$s_{i_k}^{T_{\mathbf{x},\mathbf{e}}} = \left| \frac{q(\mathbf{t}_k) - q(\mathbf{t}_{k-1})}{e_{i_k} - x_{i_k}} \right|. \tag{A10}$$

The absolute value follows (Campolongo et al., 2007), whose revised measure $\mu_j^*(\mathbf{x})$ prevents cancellation of elementary effects with opposite signs in non-monotonic models. This captures the total sensitivity of each feature, including interaction effects, rather than the net directional effect. The formal connection between SensX values and local derivatives is established in Appendix A.8.

We define the SensX value of feature $i_k$ with respect to input $\mathbf{x}$ as the mean absolute elementary effect over all grounded trajectories,

$$\widehat{s}_{\mathbf{x},i_k} = \frac{1}{n_w} \sum_{T_{\mathbf{x},\mathbf{e}} \in \mathcal{T}_{\mathbf{x}}} s_{i_k}^{T_{\mathbf{x},\mathbf{e}}}. \tag{A11}$$

The SensX vector with respect to input $\mathbf{x}$ is

$$\mathbf{s}_{\mathbf{x}} = (\widehat{s}_{\mathbf{x},1}, \ldots, \widehat{s}_{\mathbf{x},n}). \tag{A12}$$

The SensX rank vector is

$$\mathbf{r}_{\mathbf{x}} = (r_{\mathbf{x},1}, \ldots, r_{\mathbf{x},n}) \tag{A13}$$

where $r_{\mathbf{x},k}$ is the rank of $\widehat{s}_{\mathbf{x},k}$ in $\mathbf{s}_{\mathbf{x}}$, with $r_{\mathbf{x},k} = 1$ assigned to the largest SensX value. Ties are broken by feature index.

### A.6 Grouped Sensitivity Analysis

We extend SensX to allow grouping of features. This reduces the trajectory length from the number of features $D$ to the number of groups of features $G$, while preserving independent perturbations for each feature.

Let $\{I_1, \ldots, I_G\}$ be a partition of the $D$ input features into $G$ non-overlapping groups of equal size. Each feature is assigned an independent perturbation target drawn uniformly within its local perturbation bounds,

but all features within a group are perturbed simultaneously in a single trajectory step. For an input $\mathbf{x}$ and its corresponding perturbation target $\mathbf{e}$, a step at group $m$ replaces all feature values in $I_m$ with their perturbed counterparts. The grouped elementary effect is defined as:

$$s_m^{T_{\mathbf{x},\bar{\mathbf{e}}}} = \left| \frac{q(\mathbf{t}_m) - q(\mathbf{t}_{m-1})}{\|\mathbf{e}_{I_m} - \mathbf{x}_{I_m}\|_2} \right| \tag{A14}$$

where the $L_2$ norm normalizes by the magnitude of the perturbation in the subspace of the $m$-th group. As with the standard analysis, the group ordering is randomized across trajectories, and the final grouped SensX value is the mean absolute finite difference over $n_w$ trajectories.

The current implementation restricts grouped attribution to equal-sized groups because the expected $L_2$ norm of a random perturbation grows as $O(\sqrt{d})$ with group size $d$, making sensitivities across groups of different sizes incomparable. Extending the framework to unequal group sizes with appropriate normalization factors is left to future work.

## A.7 Perturbation Validation Protocol

To validate that a feature ranking concentrates model sensitivity into its highest-ranked features, we perform a top-$k$/bottom-$k$ perturbation analysis. Given a ranking of $n$ features, for each value of $k$ in a specified set, the $k$ highest-ranked (top-$k$) or $k$ lowest-ranked (bottom-$k$) features are perturbed: each selected feature is replaced by a value drawn uniformly at random within the local perturbation bounds defined by $\delta^*$, while all remaining features are held fixed. For each $k$, $N_p$ independent perturbations are generated and the QOI is recorded. If the ranking is effective, top-$k$ perturbation produces a rapid change in QOI at small $k$, while bottom-$k$ perturbation has little effect until the majority of features are perturbed.

To summarize each perturbation curve, we compute the normalized area under the curve (AUC). The $k$-axis is normalized to $[0,1]$ by dividing by the total number of features, and the QOI at each $k$ (median over $N_p$ perturbations) is divided by the unperturbed baseline prediction. For top-$k$, lower AUC indicates that the method's highest-ranked features reduce the prediction faster. For bottom-$k$, higher AUC indicates that the method's lowest-ranked features can be perturbed with less effect.

## A.8 Convergence of SensX Values

We establish what SensX estimates under progressively weaker model assumptions. Propositions 1 and 2 give exact or strong convergence results for linear and additive models at any $\delta^*$. Proposition 3 gives a general convergence result for any continuously differentiable model in the limit $\delta^* \to 0$.

**Proposition 1** (Linear models). *Let $q(\mathbf{x}) = \mathbf{w}^\top \mathbf{x} + b$ be a linear function. Then for any input $\mathbf{x}$, any $\delta^* > 0$, and any number of walks $n_w \geq 1$, the SensX value of feature $j$ is exactly $\widehat{s}_{\mathbf{x},j} = |w_j|$.*

*Proof.* Along any grounded trajectory $T_{\mathbf{x},\mathbf{e}}$, the elementary effect of feature $i_k$ at step $k$ is

$$d_{i_k}^{T_{\mathbf{x},\mathbf{e}}} = \frac{q(\mathbf{t}_k) - q(\mathbf{t}_{k-1})}{e_{i_k} - x_{i_k}} = \frac{w_{i_k}(e_{i_k} - x_{i_k})}{e_{i_k} - x_{i_k}} = w_{i_k},$$

since $\mathbf{t}_k$ and $\mathbf{t}_{k-1}$ differ only in feature $i_k$. Therefore $|d_{i_k}^{T_{\mathbf{x},\mathbf{e}}}| = |w_{i_k}|$ for every trajectory, and the mean absolute elementary effect is $\widehat{s}_{\mathbf{x},j} = |w_j|$. $\qquad\square$

**Proposition 2** (Additive models). *Let $q(\mathbf{x}) = \sum_{j=1}^n g_j(x_j)$ where each $g_j$ is continuously differentiable. Let $[a_j, b_j]$ denote the interval for feature $j$ within $\Omega(\mathbf{x}, \delta_{\mathbf{x}}^*)$, i.e., $a_j = \max\{f_j^-, x_j - \delta^* r_j\}$ and $b_j = \min\{f_j^+, x_j + \delta^* r_j\}$. Then as $n_w \to \infty$,*

$$\widehat{s}_{\mathbf{x},j} \xrightarrow{a.s.} \mathbb{E}_{e_j \sim \mathrm{Unif}[a_j,b_j]} \left[ \left| \frac{g_j(e_j) - g_j(x_j)}{e_j - x_j} \right| \right],$$

*where the expectation is over the random endpoint. Furthermore, as $\delta^* \to 0$,*

$$\widehat{s}_{\mathbf{x},j} \xrightarrow{a.s.} |g_j'(x_j)|.$$

*Proof.* For an additive model, the elementary effect of feature $j$ along any trajectory is $d_j^{T_{\mathbf{x},\mathbf{e}}} = (g_j(e_j) - g_j(x_j))/(e_j - x_j)$, which depends only on the endpoint value $e_j$ and not on the permutation order or the values of other features. Since $e_j$ is drawn uniformly from $[a_j, b_j]$ independently across walks, the SensX value $\widehat{s}_{\mathbf{x},j} = (1/n_w) \sum_w |d_j^{T_{\mathbf{x},\mathbf{e}}}|$ is a sample mean of i.i.d. random variables and converges almost surely by the strong law of large numbers. As $\delta^* \to 0$, the interval $[a_j, b_j]$ contracts to $\{x_j\}$ and the difference quotient converges to $g_j'(x_j)$ by differentiability. $\square$

**Proposition 3** (General smooth models). *Let $q : \mathbb{R}^n \to \mathbb{R}$ be continuously differentiable in a neighborhood of $\mathbf{x}$. Then for each feature $j$ and any fixed $n_w \geq 1$, as $\delta^* \to 0$, every elementary effect $d_j^{T_{\mathbf{x},\mathbf{e}}}$ converges to $\partial q / \partial x_j(\mathbf{x})$ uniformly over all trajectories and endpoints, and consequently*

$$\widehat{s}_{\mathbf{x},j} \to \left| \frac{\partial q}{\partial x_j}(\mathbf{x}) \right|.$$

*Proof.* Consider the elementary effect of feature $j$ at step $k$ of a grounded trajectory $T_{\mathbf{x},\mathbf{e}}$, where feature $j$ is the $k$-th feature to be perturbed. Since the trajectory is anchored at $\mathbf{x}$ and feature $j$ has not yet been changed at step $k-1$, we have $t_{k-1,j} = x_j$. The points $\mathbf{t}_{k-1}$ and $\mathbf{t}_k$ differ only in coordinate $j$, with $t_{k,j} = e_j$. By the mean value theorem applied to the function $z \mapsto q(t_{k-1,1}, \ldots, z, \ldots, t_{k-1,n})$, there exists $\xi_j$ between $x_j$ and $e_j$ such that

$$q(\mathbf{t}_k) - q(\mathbf{t}_{k-1}) = \frac{\partial q}{\partial x_j}(t_{k-1,1}, \ldots, \xi_j, \ldots, t_{k-1,n}) \cdot (e_j - x_j).$$

Since the denominator of the elementary effect (Equation A10) is also $e_j - x_j$, we obtain

$$d_j^{T_{\mathbf{x},\mathbf{e}}} = \frac{\partial q}{\partial x_j}(t_{k-1,1}, \ldots, \xi_j, \ldots, t_{k-1,n}),$$

where $\xi_j$ lies between $x_j$ and $e_j$, and the remaining coordinates $t_{k-1,i}$ for $i \neq j$ equal either $x_i$ (if feature $i$ has not yet been perturbed) or $e_i$ (if it has). As $\delta^* \to 0$, all endpoints satisfy $\mathbf{e} \to \mathbf{x}$, so all intermediate trajectory points satisfy $\mathbf{t}_{k-1} \to \mathbf{x}$ and $\xi_j \to x_j$. By continuity of the partial derivative,

$$d_j^{T_{\mathbf{x},\mathbf{e}}} \to \frac{\partial q}{\partial x_j}(\mathbf{x})$$

for every trajectory, every permutation, and every endpoint. Since absolute value is continuous, $|d_j^{T_{\mathbf{x},\mathbf{e}}}| \to |\partial_j q(\mathbf{x})|$ uniformly over all trajectories, and the mean $\widehat{s}_{\mathbf{x},j} \to |\partial_j q(\mathbf{x})|$. $\square$

### A.9 Morris, DGSM, and the Formal Context for the Anchored Estimand

An important distinction between SensX and the standard Morris method is the anchoring of trajectories at $\mathbf{x}$. In the standard method, trajectories begin at random points uniformly distributed within the domain. Morris (1991) proposed the signed mean $\mu_j$ and standard deviation $\sigma_j$ of the elementary effects of feature $j$ across trajectories as sensitivity measures: large $|\mu_j|$ indicates overall influence on the output; large $\sigma_j$ indicates nonlinearity or interactions, because the elementary effect of feature $j$ varies across trajectories when other features modulate its impact. Campolongo et al. (2007) introduced $\mu_j^*$, the mean of the absolute elementary effects, to resolve sign cancellation in nonmonotonic models, where positive and negative effects cancel in $\mu_j$ and render important factors invisible, and demonstrated empirically that $\mu_j^*$ is approximately proportional to $S_{Ti}$ (Sobol, 2001), the total-order Sobol sensitivity index. This empirical observation has no formal proof; the closest rigorous result is as follows. The formal connection between Morris-style measures and model derivatives was identified by Sobol' and Kucherenko (2009), who introduced Derivative-based

Global Sensitivity Measures (DGSM), $\nu_j = \int_{[0,1]^n} (\partial q/\partial x_j)^2 \, d\mathbf{x}$, showed that the Morris $\mu_j^*$ approximates $\mu_j = \int_{[0,1]^n} |\partial q/\partial x_j| \, d\mathbf{x}$ in the small-step limit, and proved via the Poincaré inequality that for uniform input distributions,

$$S_{Ti} \ \leq \ \frac{\pi^2}{4} \cdot \frac{\nu_j}{\text{Var}(q)},$$

where $C_j = \pi^2/4$ is the Poincaré constant for the uniform distribution. Lamboni et al. (2013) extended this inequality to the broader class of Boltzmann probability measures using a general Poincaré inequality. Note that this bound applies to the standard Morris method in its small-step limit (i.e., the squared-derivative integral $\nu_j$, not to $\mu_j^*$ directly, and not to anchored SensX when $\delta^*$ is bounded away from zero).

The broader lineage connecting sensitivity methods to variance decomposition is instructive. Cukier et al. (1973) introduced FAST, showing that Fourier spectral coefficients of the model output estimate the variance contributions $S_j$ when inputs are driven by incommensurate oscillatory functions; Cukier et al. (1978) further developed approximation results with computable error bounds under smoothness conditions. Saltelli and Bolado (1998) proved that the FAST spectral estimator and the Sobol' first-order index $S_j = \text{Var}(\mathbb{E}[q \mid x_j])/\text{Var}(q)$ estimate the same ANOVA main-effect quantity. The asymmetry with Morris is informative: FAST has a formal approximation theory for $S_j$ with convergence guarantees, whereas the Morris $\mu_j^*$ has only an empirical observation of approximate proportionality with $S_{Ti}$ (Campolongo et al., 2007) and no analogous formal theory. The DGSM result partially closes this gap, but only in the $\delta^* \to 0$ limit and only for the global (domain-wide) method: the small-step limit in DGSM is precisely the $\delta^* \to 0$ limit, since shrinking the trajectory step size to zero is equivalent to shrinking the neighborhood from which endpoints are drawn. In this limit, Proposition 3 establishes the pointwise analogue for anchored SensX ($\widehat{s}_{\mathbf{x},j} \to |\partial_j q(\mathbf{x})|$), and the Poincaré-based bound toward $S_{Ti}$ then applies to the domain-wide aggregate $\nu_j = \int (\partial_j q)^2 \, d\mathbf{x}$, not to any single anchored computation. In the operative regime where $\delta^*$ is bounded away from zero, none of these limit results apply and the analytical characterization of the anchored $\mu_j^*$ estimand remains an open problem (Appendix A.11).

### A.10 Finite-Sample Stability of the Characteristic Perturbation Scale

We establish the stability of the estimated $\delta_{\mathbf{x}}^*$ through three complementary results: monotonicity of the population saturation criterion, concentration of the estimated criterion under finite sampling, and a bound on discretization error under grid refinement.

**Notation.** For a given input $\mathbf{x}$ and perturbation scale $\delta \in \Delta$, let $\overline{D}(\delta)$ and $\underline{D}(\delta)$ denote the population 99th and 1st percentiles of the QOI distribution $D(\mathbf{x}, \delta, n_s)$, and let $\widehat{\overline{D}}(\delta)$ and $\widehat{\underline{D}}(\delta)$ denote their empirical counterparts from $n_s$ i.i.d. uniform samples. The population saturation criterion is

$$C(\delta') = \max_{\delta \in \Delta:\, \delta > \delta'} \overline{D}(\delta) - \min_{\delta \in \Delta:\, \delta > \delta'} \underline{D}(\delta),$$

and $\delta_{\mathbf{x}}^*$ is the smallest $\delta' \in \Delta$ such that $C(\delta') < \tau_a$ (Equation A7).

**Proposition 4** (Monotonicity of the saturation criterion). *For any $\delta_1' < \delta_2'$ in $\Delta$, $C(\delta_1') \geq C(\delta_2')$.*

*Proof.* The set $\{\delta \in \Delta : \delta > \delta_2'\} \subseteq \{\delta \in \Delta : \delta > \delta_1'\}$. The maximum over a subset cannot exceed the maximum over the superset, so $\max_{\delta > \delta_2'} \overline{D}(\delta) \leq \max_{\delta > \delta_1'} \overline{D}(\delta)$. Similarly, the minimum over a subset cannot be less than the minimum over the superset, so $\min_{\delta > \delta_2'} \underline{D}(\delta) \geq \min_{\delta > \delta_1'} \underline{D}(\delta)$. Subtracting gives $C(\delta_2') \leq C(\delta_1')$. $\square$

Proposition 4 implies that the set $\{\delta' \in \Delta : C(\delta') < \tau_a\}$ is an upper segment of $\Delta$: once the criterion drops below $\tau_a$ at some $\delta'$, it remains below $\tau_a$ for all larger $\delta'$. The population $\delta_{\mathbf{x}}^*$ is the infimum of this segment. This structure ensures that the calibration step identifies a unique transition point rather than oscillating between eligible scales.

**Estimation consistency.** Both percentiles at each $\delta \in \Delta$ are estimated from $n_s$ i.i.d. uniform samples. By the Dvoretzky–Kiefer–Wolfowitz (DKW) inequality, the empirical CDF satisfies

$$\Pr\left(\sup_t \left| \hat{F}_{n_s}(t) - F(t) \right| > \varepsilon_0 \right) \leq 2e^{-2n_s \varepsilon_0^2}$$

for any $\varepsilon_0 > 0$. To translate CDF accuracy into quantile accuracy, the density at the relevant percentiles is required. Let $f_{\min} > 0$ denote the infimum of the QOI density at the 1st and 99th percentiles across all $\delta \in \Delta$. When the QOI distribution has positive density at these percentiles (generically satisfied when $q$ is continuous and the perturbation distribution is absolutely continuous), a CDF deviation of at most $\varepsilon_0$ implies that the corresponding empirical percentile deviates from its population value by at most $\varepsilon_0 / f_{\min}$ (Serfling, 1980, Chapter 2). Setting $\varepsilon_0 = \varepsilon \cdot f_{\min}$ to achieve quantile accuracy $\varepsilon$ and applying a union bound over the $2|\Delta|$ percentile estimates (two per grid point) gives

$$\Pr(\text{all } 2|\Delta| \text{ percentile estimates within } \varepsilon \text{ of population values}) \geq 1 - 4|\Delta| \, e^{-2n_s \varepsilon^2 f_{\min}^2}.$$

The saturation criterion $C(\delta')$ involves a maximum and a minimum over subsets of these percentile estimates. Since the maximum (resp. minimum) over a finite set shifts by at most $\varepsilon$ when each element shifts by at most $\varepsilon$, the estimated saturation criterion $\hat{C}(\delta')$ deviates from $C(\delta')$ by at most $2\varepsilon$. Therefore, the estimated $\delta_{\mathbf{x}}^*$ equals the population $\delta_{\mathbf{x}}^*$ once $2\varepsilon < \gamma$, where

$$\gamma = \min_{\delta' \in \Delta} |C(\delta') - \tau_a|$$

is the margin of the population criterion from the threshold. This requires the non-degeneracy condition $\gamma > 0$, i.e., no grid point has its population criterion exactly at $\tau_a$. For continuous models with absolutely continuous perturbation distributions, $C(\delta')$ is a continuous function of $\delta'$ and $\tau_a$ is user-specified, so exact equality at any finite set of grid points is a measure-zero event. Substituting $\varepsilon = \gamma/2$ yields the sufficient sample size

$$n_s \geq \frac{\log(4|\Delta|/\alpha)}{2(\gamma f_{\min}/2)^2} = \frac{2\log(4|\Delta|/\alpha)}{\gamma^2 f_{\min}^2}$$

for the estimated $\delta_{\mathbf{x}}^*$ to equal the population value with probability at least $1 - \alpha$. Neither $\gamma$ nor $f_{\min}$ is known a priori, but both are positive under the stated regularity conditions, and the bound confirms that $n_s$ scales as $O(\gamma^{-2} f_{\min}^{-2} \log |\Delta|)$.

**Discretization error.** The estimation analysis above addresses how accurately $n_s$ samples recover the population criterion at each fixed grid point. A distinct source of error arises because the true continuous minimizer $\delta_{\text{cont}}^*$ may lie strictly between two consecutive grid points.

**Proposition 5** (Grid refinement convergence). *Let $C(\delta')$ be the population saturation criterion, which is non-increasing by the same subset argument as Proposition 4 (applied to the continuous domain $[0,1]$). Let $\delta_{\text{cont}}^*$ denote the smallest $\delta'$ in $[0,1]$ such that $C(\delta') < \tau_a$, and let $\delta_\Delta^*$ denote the smallest $\delta' \in \Delta$ such that $C(\delta') < \tau_a$. If $C$ is strictly decreasing in a neighborhood of $\delta_{\text{cont}}^*$, then*

$$0 \leq \delta_\Delta^* - \delta_{\text{cont}}^* \leq h,$$

*where $h = \delta_{i+1} - \delta_i$ is the spacing of the grid interval $[\delta_i, \delta_{i+1}]$ that contains $\delta_{\text{cont}}^*$.*

*Proof.* Since $C$ is non-increasing, $C(\delta') < \tau_a$ for all $\delta' > \delta_{\text{cont}}^*$. The grid point $\delta_\Delta^*$ is the smallest element of $\Delta$ in this region, which is $\delta_{i+1}$, the right endpoint of the grid interval containing $\delta_{\text{cont}}^*$. Therefore $\delta_\Delta^* - \delta_{\text{cont}}^* \leq \delta_{i+1} - \delta_i = h$. The lower bound $\delta_\Delta^* \geq \delta_{\text{cont}}^*$ follows because $C(\delta') \geq \tau_a$ for all $\delta' < \delta_{\text{cont}}^*$, and therefore for any grid point below $\delta_{\text{cont}}^*$. $\square$

Proposition 5 establishes that the discretization error vanishes as the grid is refined, with the rate controlled by local grid spacing near the transition. The effect of grid type and resolution on $\delta^*$ is demonstrated concretely in Appendix B (Figure B1) for the single-cell classifiers, where the sensitivity transition occurs near zero: a log-spaced grid concentrates resolution near the transition, reducing $h$ where it matters, and increasing $k$ (the number of grid points) produces estimates that track the transition more precisely, consistent with Proposition 5.

### A.11 Limitations of the Theoretical Characterization

The propositions above characterize SensX values under idealized conditions or in asymptotic limits. We state explicitly what is *not* established.

**$\delta^*$ bounded away from zero.** Propositions 1–2 characterize SensX values for all $\delta^*$ under structural model assumptions (linearity or additivity). The additive case escapes the difficulty because elementary effects depend only on the feature's own endpoint, not on permutation order or other features. For a general non-additive model when $\delta^*$ is bounded away from zero, the SensX value $\widehat{s}_{\mathbf{x},j}$ is the sample mean of elementary effects whose distribution depends jointly on the model's interaction structure, the permutation order, and the endpoint distribution over $\Omega(\mathbf{x}, \delta_{\mathbf{x}}^*)$. This distribution does not simplify to a known analytic functional in general. Proposition 3 establishes only that it approaches $|\partial_j q(\mathbf{x})|$ as $\delta^* \to 0$.

**No formal connection to total Sobol index in the anchored setting.** $S_{Ti}$ is defined as an integral over the full input domain and has no analogue at a single point. SensX anchors trajectories at $\mathbf{x}$ by design: the goal is local attribution, and $\mathbf{x}$ serves as the reference with respect to which all elementary effects are computed. Campolongo et al. (2007) introduced $\mu_j^*(\mathbf{x})$ to resolve sign cancellation in $\mu_j(\mathbf{x})$ for nonmonotonic models, and demonstrated empirically that $\mu_j^*$ is approximately proportional to $S_{Ti}$ in the global setting. This motivates using $\mu_j^*$ over $\mu_j$, but $\mu_j^*$ and $S_{Ti}$ measure different mathematical quantities and no theorem establishing a formal relationship between them exists. In the $\delta^* \to 0$ limit, Proposition 3 shows the estimand converges to the local partial derivative $|\partial_j q(\mathbf{x})|$, a first-order quantity with no formal connection to $S_{Ti}$.

The Lebesgue Differentiation Theorem provides a bridge: the DGSM integral $\int |\partial_j q| \, d\mathbf{x}$, which sits in the inequality chain toward $S_{Ti}$, equals the average of the pointwise limits $|\partial_j q(\mathbf{x})|$ over inputs drawn from $\Omega^g$ at almost every point. The formal DGSM connection to $S_{Ti}$ is therefore recoverable by aggregating SensX values across many inputs, recovering the standard Morris method, but is not available from any single anchored computation. The perturbation validation protocol (Appendix A.7) provides a model-specific empirical check that rankings are meaningful for the specific input being explained.

**No non-asymptotic concentration bound on rankings.** The almost-sure convergence in Proposition 2 follows from the strong law of large numbers for i.i.d. random variables, with a $O(1/\sqrt{n_w})$ rate via the central limit theorem under finite second-moment conditions (on the elementary effect distribution). No analogous rate is established for the convergence of the *ranking* of SensX values, which depends on the gaps between features' expected elementary effects in ways that are model-specific. The empirical stability of rankings with increasing $n_w$ is documented in Figure E1, but a theoretical analysis of ranking convergence is outside the scope of this work.

**Type II error and underestimation of interaction effects.** Two distinct Type II failure modes are relevant to SensX. The first is specific to local attribution: even at its limit $|\partial_j q(\mathbf{x})|$, SensX will assign a near-zero value to a feature $j$ that has a negligible partial derivative at $\mathbf{x}$ even if it has a substantial effect elsewhere in the domain. The $\delta^*$ calibration partially mitigates this by selecting a neighborhood where the model is actively sensitive, but it cannot detect features that are important only outside $\Omega(\mathbf{x}, \delta_{\mathbf{x}}^*)$. When $\delta_{\mathbf{x}}^* = 0$, the diagnostic output of SensX flags this situation explicitly rather than producing misleading attributions. The second failure mode is inherited from the Morris $\mu_j^*$ statistic itself and operates when $\delta^*$ is bounded away from zero: Campolongo et al. (2007) noted, and Confalonieri et al. (2010) and Hsieh et al. (2018) confirmed across multiple domains, that $\mu_j^*$ tends to underestimate the importance of features involved in strong interactions, because when a feature's effect is highly dependent on the values of other features, individual elementary effects may be large in magnitude but variable in direction, and their absolute values averaged over a finite number of walks may not fully reflect the total sensitivity. This is a tendency under limited sampling, not a guarantee of failure, and it applies to both the standard Morris method and anchored SensX. The perturbation validation protocol (Appendix A.7) provides a model-specific empirical check that can detect such cases, since a factor whose $\mu_j^*$ is depressed by sign cancellation will still show a large output change when perturbed in the top-$k$ analysis.

## B    Perturbation Factor Grid Selection

For synthetic benchmarks, $\delta^*$ was close to 1 for all inputs, so a uniform partition of $[0, 1]$ provided sufficient resolution and grid refinement did not change the result. For the image-based tasks of vision transformers and spatial transcriptomics, a uniform grid was likewise adequate, as $\delta^*$ fell well above the minimum grid spacing.

For single-cell transcriptomics, the prediction envelope changed rapidly at small perturbation magnitudes. Replacing the uniform grid with a log-spaced $\Delta$ ($\delta \in [10^{-4}, 1]$) concentrated resolution at the fine scales where the model's sensitivity was steepest and yielded stable estimates of $\delta^*$. When a grid has no resolution near zero and all points fall in the saturated plateau, the saturation criterion is never met and $\delta^* = 0$. This is an explicit diagnostic: rather than returning a misleading non-zero value, SensX flags the situation, signaling that the grid should be refined or that the model is genuinely insensitive to perturbations at any scale within $\Omega^g$.

Figure B1 shows the joint sensitivity of $\delta^*$ to $\tau_a$, $k$, and $n_s$ for a single natural killer cell under both grid types. With a linspace grid, $\delta^* = 0$ across all $(k, \tau_a, n_s)$ combinations: all grid points fall in the saturated plateau above the sensitivity transition, the saturation criterion is never met, and the method correctly flags the situation rather than returning a misleading non-zero value. With a geomspace grid, $\delta^*$ is stable and non-zero across all $k$ and $n_s$ values, confirming that log-spaced resolution near zero recovers the sensitivity transition. $\delta^*$ shows greater sensitivity to $k$ at small $\tau_a$ than at larger values: when $\tau_a$ is large, features driving substantial changes in predicted probability dominate the saturation criterion and $\delta^*$ is stable regardless of grid resolution. When $\tau_a$ is small, subtler output variations are deemed significant and $\delta^*$ is more susceptible to grid placement effects. The near-identical positions across all $n_s$ values within each $k$ group confirm that $n_s$ has no effect on $\delta^*$ for this classifier.

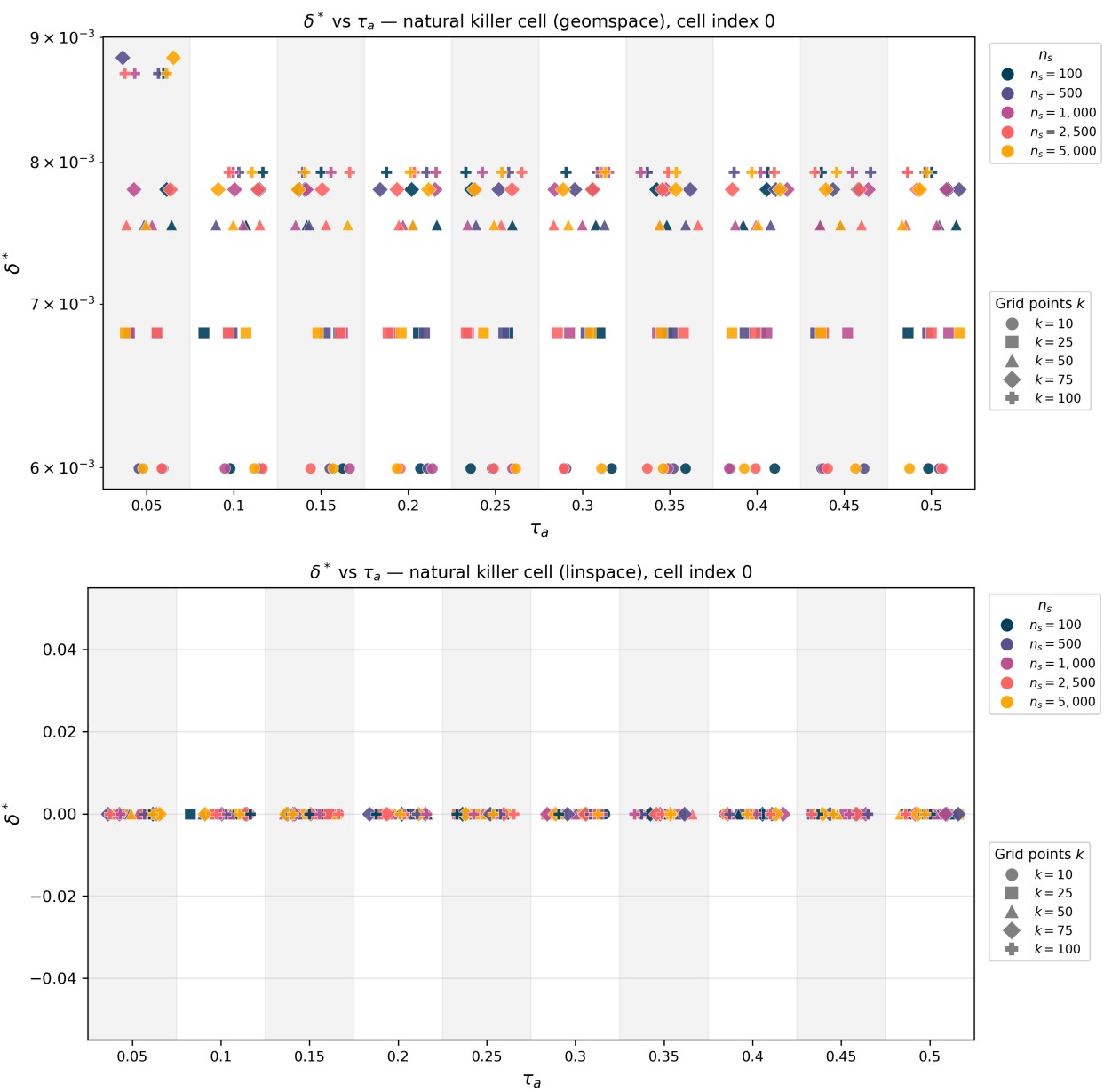

Figure B1: **Joint sensitivity of $\delta^*$ to $\tau_a$, grid resolution, and grid type for the natural killer cell classifier (single cell).** Each point shows $\delta^*$ for one combination of stability threshold $\tau_a$ (x-axis, categorical), number of grid points $k$ (marker shape), and Monte Carlo sample size $n_s$ (color). Top: log-spaced (geomspace) grid; bottom: uniform (linspace) grid. With a linspace grid, $\delta^* = 0$ for all combinations because all grid points fall in the saturated plateau above the sensitivity transition, and the method flags this explicitly rather than returning a misleading value. With a geomspace grid, $\delta^*$ is stable and non-zero across all $k$ and $n_s$, with greater variation across $k$ at small $\tau_a$ than at larger values. The near-identical positions across $n_s$ colors within each $k$ group confirm that $n_s$ has no effect on $\delta^*$.

To assess whether this stability holds across the population, we computed $n_s^*$, the smallest $n_s$ at which $\delta^*$ converges to its value at $n_s = 5{,}000$, for each of the 1,000 high-confidence natural killer cells across all $(k, \tau_a)$ combinations. Figure B2 shows the distribution of $n_s^*$ as a heatmap: rows are $n_s^*$ levels (100–5,000), columns are $\tau_a$ values, and each cell shows the percentage of cells at that level, separately for each $k$ value. For

$k = 10$, essentially 100% of cells converge at $n_s = 100$ across all $\tau_a$ values. For larger $k$, 94–99% of cells converge at $n_s = 100$, with the remainder converging by $n_s = 500$ in all but a negligible fraction of cases. The tail fraction grows monotonically with $k$: finer grids place more points near the sensitivity transition, increasing the probability that $\delta^*$ lands close to a grid boundary where small estimation noise can cause a jump to a neighboring grid value. By contrast, $\tau_a$ has no systematic effect on the convergence distribution, and no trend is visible across columns within any $k$ panel. This decoupling is consistent with the two findings measuring different quantities: the single-cell result concerns grid placement sensitivity for a fixed cell, while the population result concerns whether individual cells happen to sit near a grid boundary. Across all $(k, \tau_a)$ combinations, $n_s = 100$ is sufficient for 97.5% of natural killer cells. Figure B3 shows the same analysis for all 14 cell types; the pattern is consistent, with $n_s = 100$ dominating across all cell types and the tail fraction remaining below 6% in all cases.

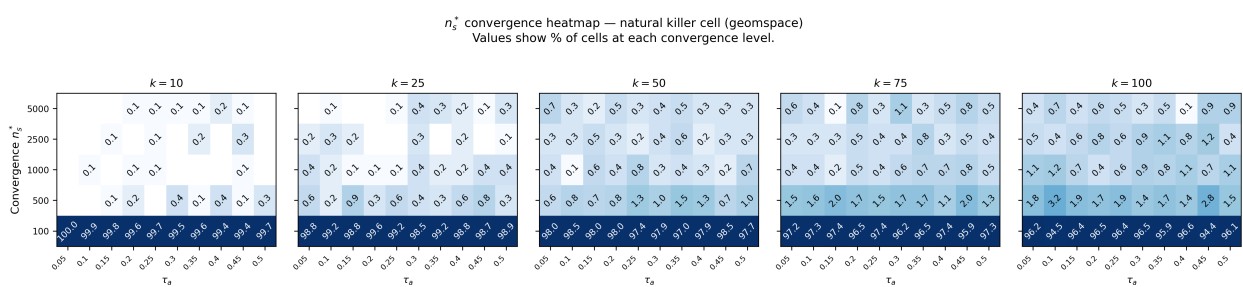

Figure B2: **Population-level $n_s^*$ convergence for the natural killer cell classifier (log-spaced grid, 1,000 cells).** Each cell shows the percentage of cells whose $\delta^*$ first converges at that $n_s^*$ level (rows), for a given $\tau_a$ value (columns) and grid resolution $k$ (panels). $n_s^* = 100$ dominates across all $k$ and $\tau_a$ values, with the fraction requiring higher $n_s$ growing modestly with $k$ but remaining below 6%. The absence of a clear $\tau_a$ pattern in the convergence distribution confirms that the need for more samples is not driven by the threshold choice, but by whether individual cells happen to sit near a grid boundary.

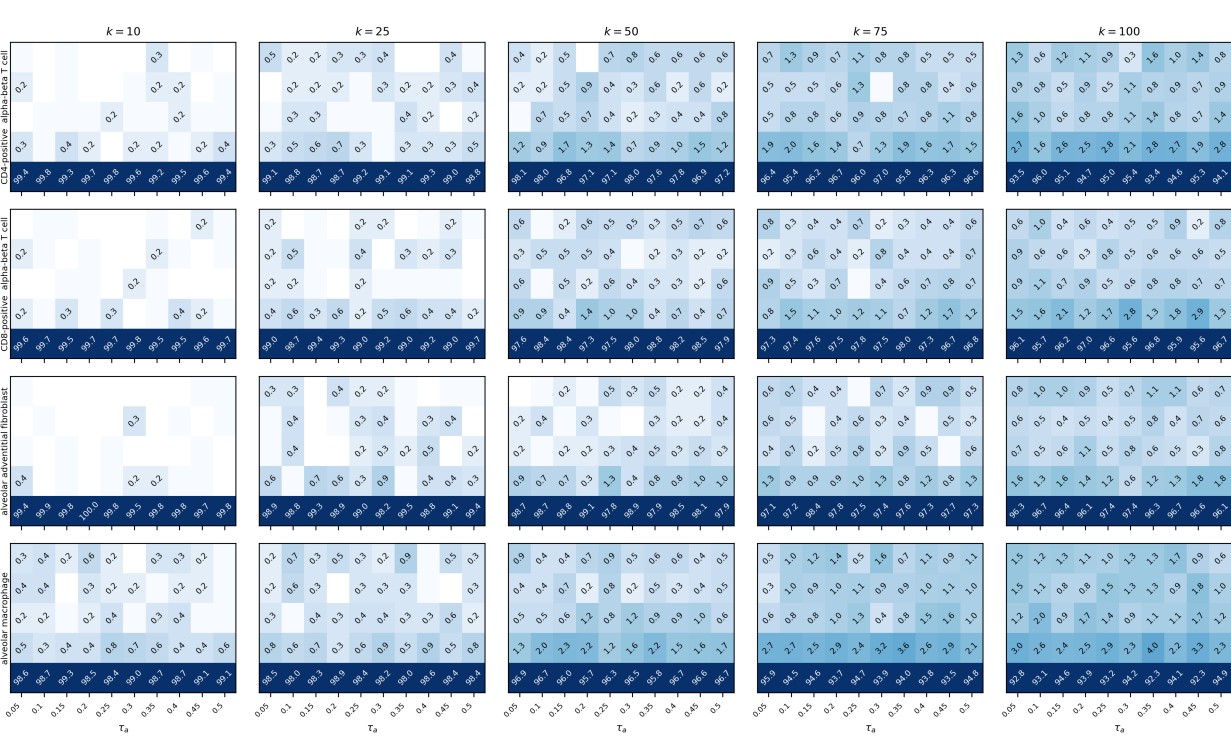

Figure B3: **Population-level $n_s^*$ convergence across all 14 cell types (log-spaced grid, part 1 of 4).** Same layout as Figure B2, with one row per cell type. The pattern is consistent across all cell types: $n_s = 100$ is sufficient for the vast majority of cells, and the tail fraction remains below 6% in all cases. Continued in Figures B4–B6.

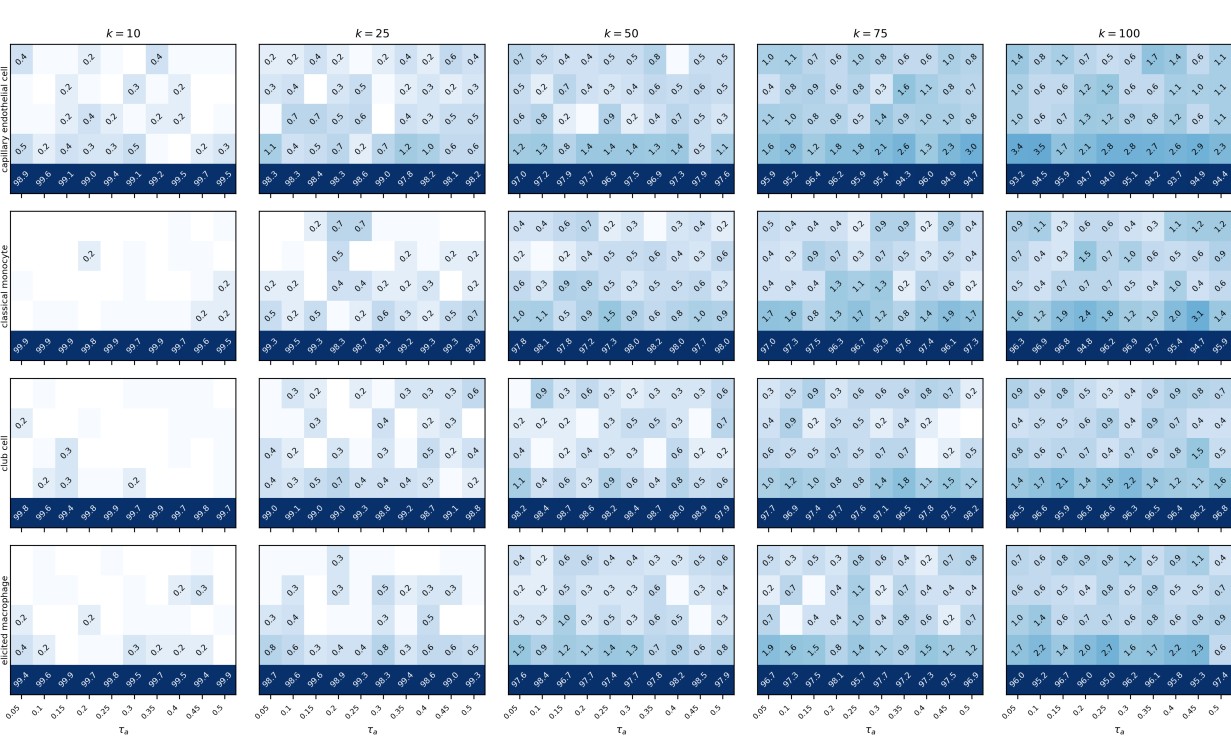

Figure B4: **Population-level $n_s^*$ convergence across all 14 cell types (log-spaced grid, part 2 of 4).** Continuation of Figure B3.

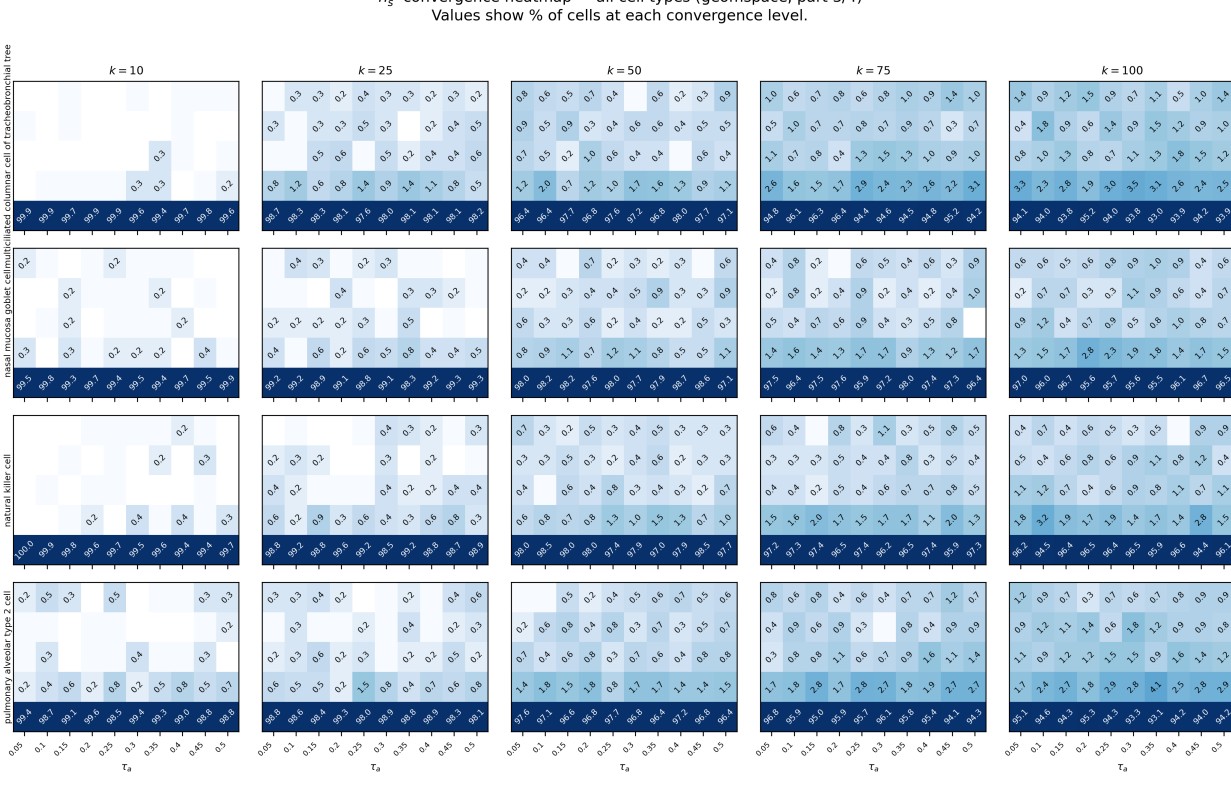

Figure B5: **Population-level $n_s^*$ convergence across all 14 cell types (log-spaced grid, part 3 of 4).** Continuation of Figure B3.

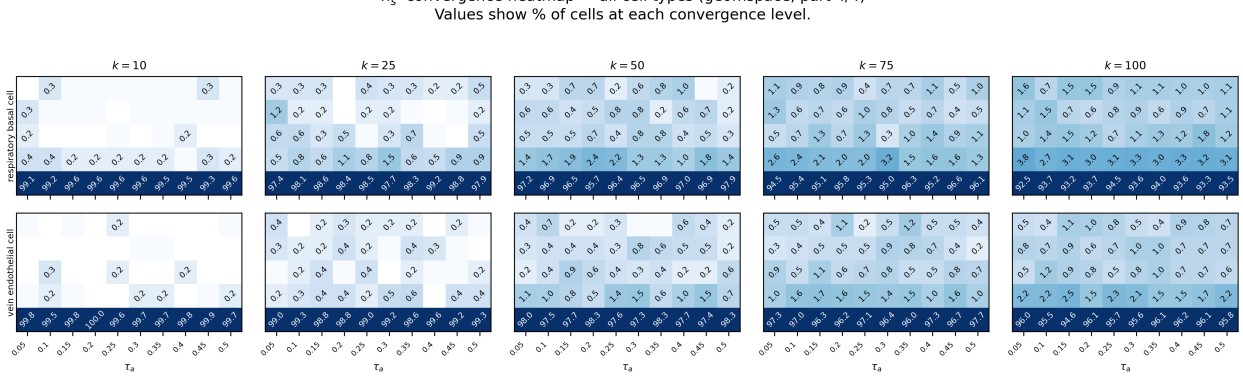

Figure B6: **Population-level $n_s^*$ convergence across all 14 cell types (log-spaced grid, part 4 of 4).** Continuation of Figure B3.

In general, we recommend starting with a uniform grid and switching to log-spaced $\Delta$ if $\delta^*$ clusters near the grid's lower bound across inputs. The perturbation validation protocol (Appendix A.7) provides the ultimate check on whether the resulting rankings are meaningful regardless of grid choice.

## C    Future Extensions

The uniform random sampling in the current implementation could be replaced by precomputed low-discrepancy quasi-random sequences (e.g., Sobol sequences) to improve convergence rates without additional computational overhead. This would connect SensX's sampling strategy to the quasi-Monte Carlo literature used in classical sensitivity analysis and in the Sobol-based attribution method of Fel et al. (2021).

The current implementation restricts grouped attribution to equal-sized groups (Appendix A.6). Extending the framework to unequal group sizes with appropriate normalization factors would enable hierarchical attribution across natural grouping structures of varying granularity.

# D   Supplementary Figures

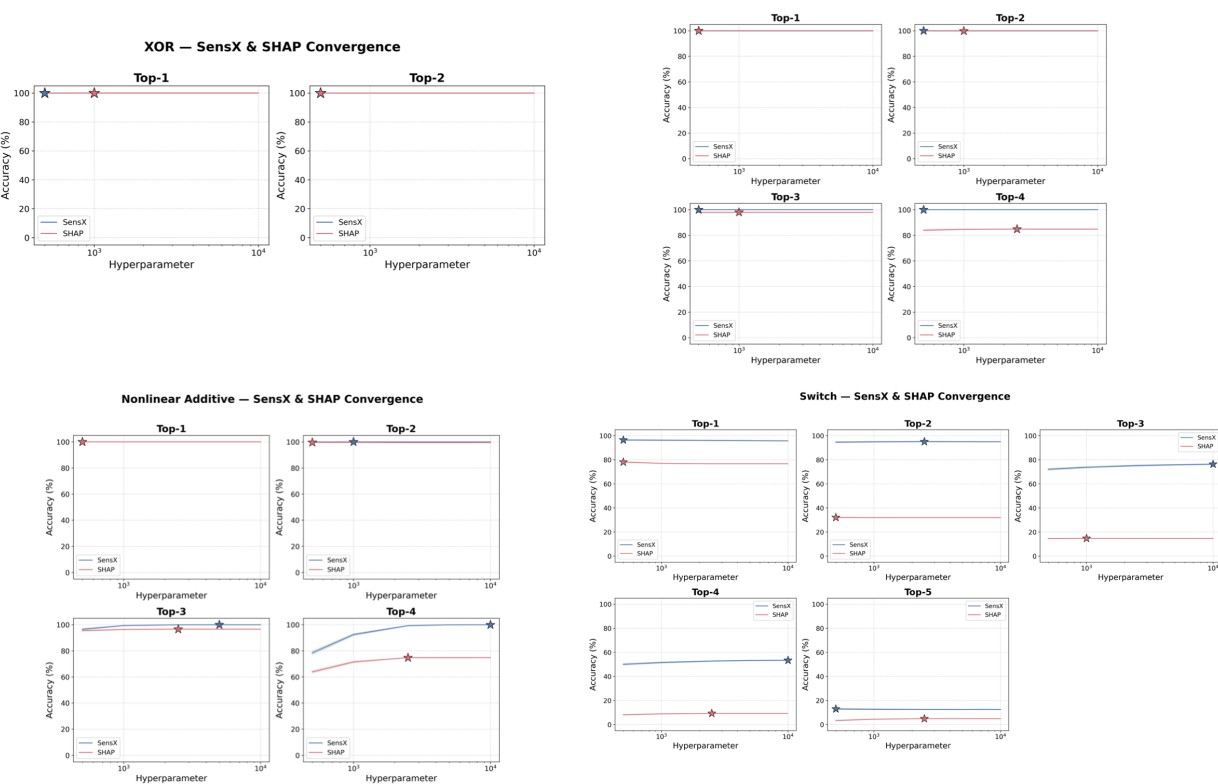

Figure D1: **SensX and KernelSHAP convergence across hyperparameters on synthetic benchmarks.** Mean top-$k$ accuracy $\pm$ 1 standard deviation (100 repetitions per configuration) as a function of each method's hyperparameter for **(A)** XOR ($k = 1, 2$), **(B)** Orange Skin ($k = 1$–4), **(C)** Nonlinear Additive ($k = 1$–4), and **(D)** Switch ($k = 1$–5). Stars mark the best-performing hyperparameter for each method, corresponding to the values reported in the bar plots. Both methods are largely insensitive to hyperparameter choice: accuracy curves are flat across the range tested, with the exception of SensX on Nonlinear Additive at top-4, where accuracy increases with the number of walks.

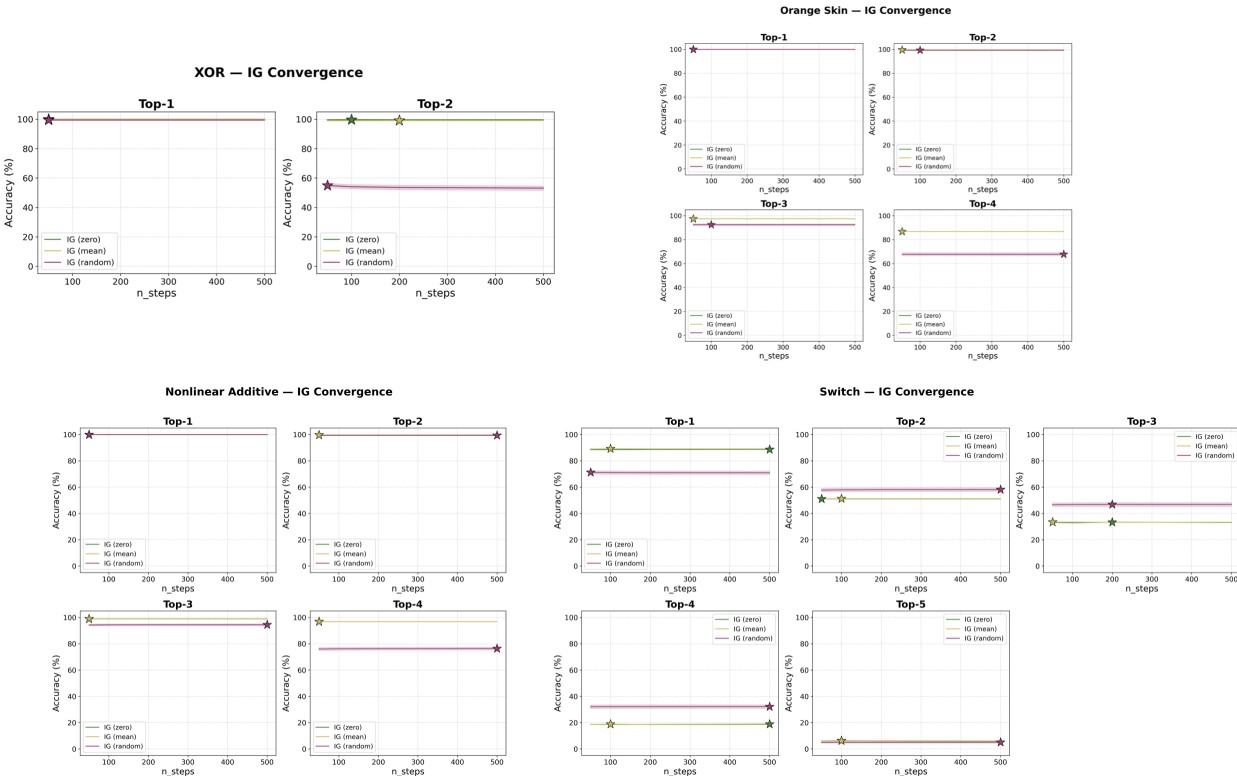

Figure D2: **Integrated gradients convergence across integration steps on synthetic benchmarks.** Mean top-$k$ accuracy as a function of the number of integration steps $n_{\text{steps}}$ for three baseline choices (zero vector (green), training-set mean (yellow), and random training sample (pink)) on **(A)** XOR ($k = 1, 2$), **(B)** Orange Skin ($k = $ 1–4), **(C)** Nonlinear Additive ($k = $ 1–4), and **(D)** Switch ($k = $ 1–5). Stars mark the best-performing configuration for each baseline type. Accuracy is largely insensitive to $n_{\text{steps}}$ across all datasets, indicating that the path integral is well-approximated even at 50 steps. Performance differences are driven by baseline choice rather than integration resolution, with the random baseline consistently matching or exceeding the deterministic baselines on the Switch dataset. Error bands for the random baseline show $\pm$ 1 standard deviation across 100 runs, while zero and mean baselines are deterministic.

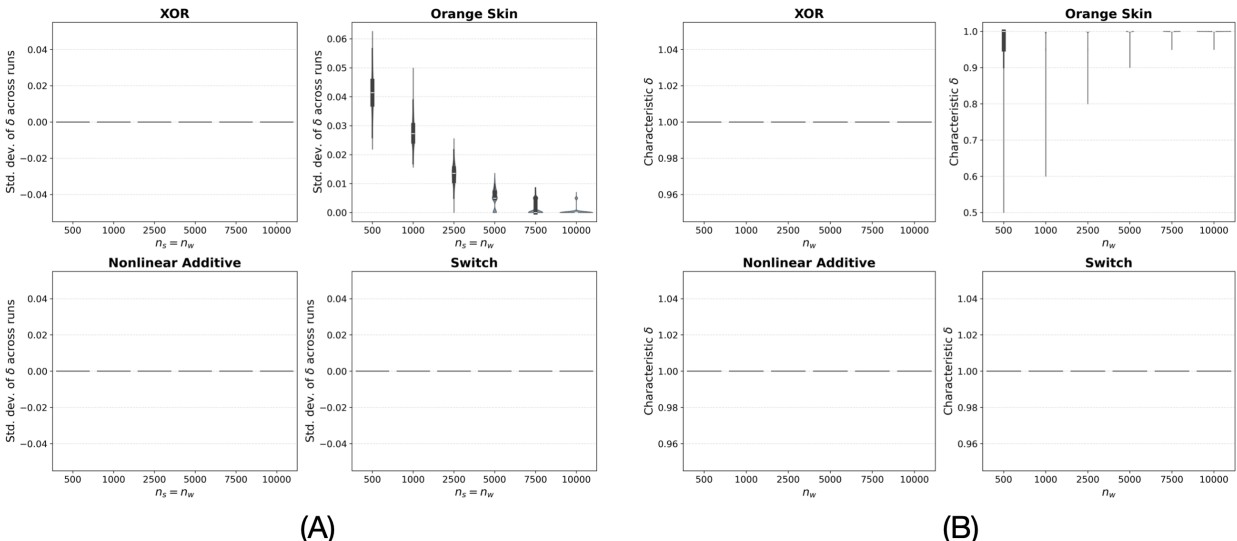

Figure D3: **Characteristic perturbation scale across synthetic datasets.** (A) Per-sample standard deviation of $\delta^*$ across 100 independent runs as a function of the number of samples ($n_s = n_w$). Only the Orange Skin dataset exhibits non-trivial variability, which decreases with increasing $n_w$. (B) Distribution of $\delta^*$ values across all samples and runs at each $n_w$. On the XOR, Nonlinear Additive, and Switch datasets, $\delta^*$ equals 1 for all inputs regardless of $n_w$. On the Orange Skin dataset, $\delta^*$ is concentrated near 1 but shows a tail of lower values that narrows as $n_w$ increases.

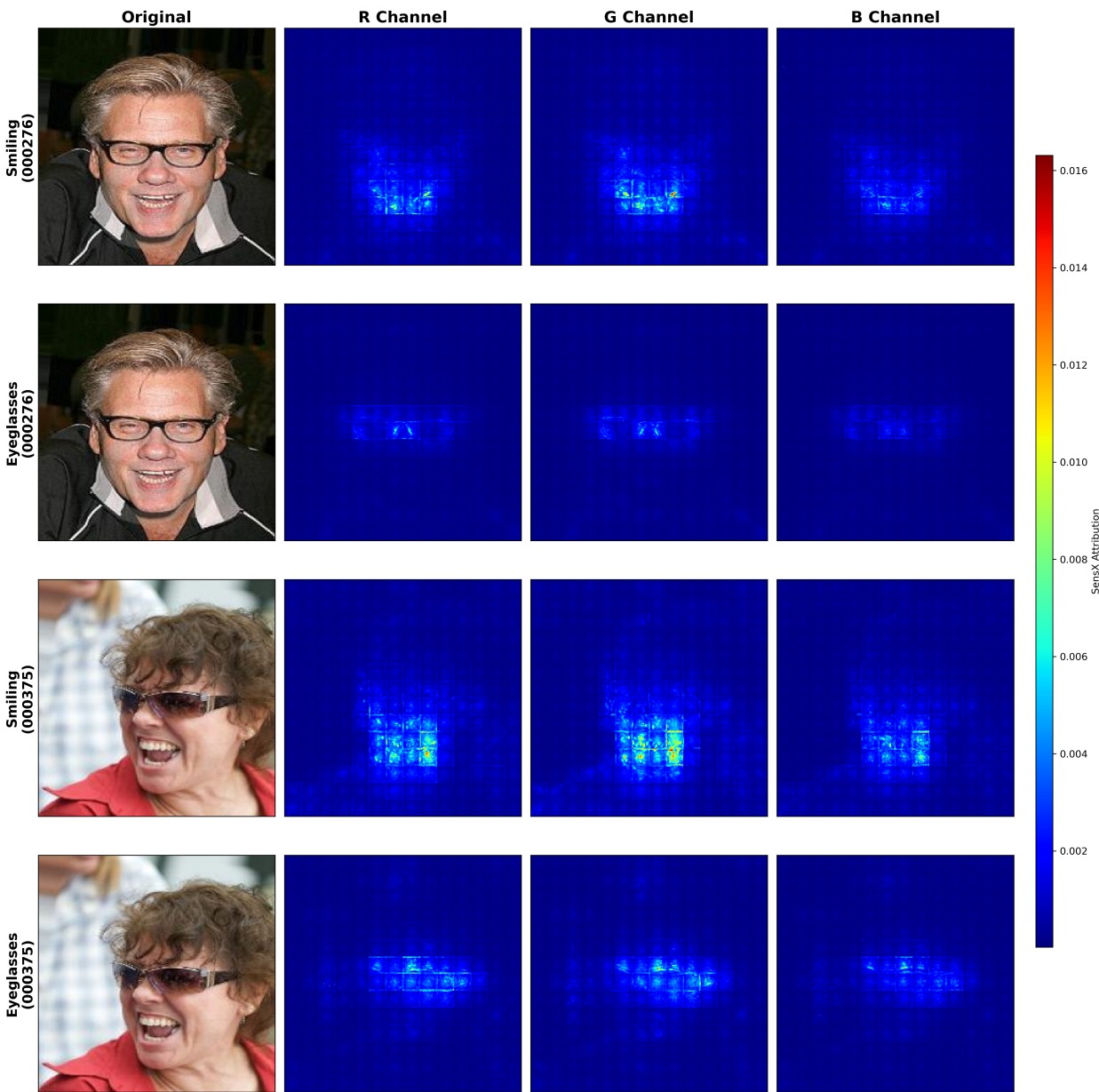

Figure D4: **Per-channel SensX attribution heatmaps for ViT models.** Each row corresponds to one (image, model) pair. Columns show the original image followed by SensX attribution values for the red, green, and blue channels independently. All heatmaps share a common color scale. Attribution patterns are consistent across color channels, confirming that the three channels at a given pixel location tend to receive similar attribution values. Grid artifacts aligned with the ViT's $16 \times 16$ patch tokenization are visible across all panels.

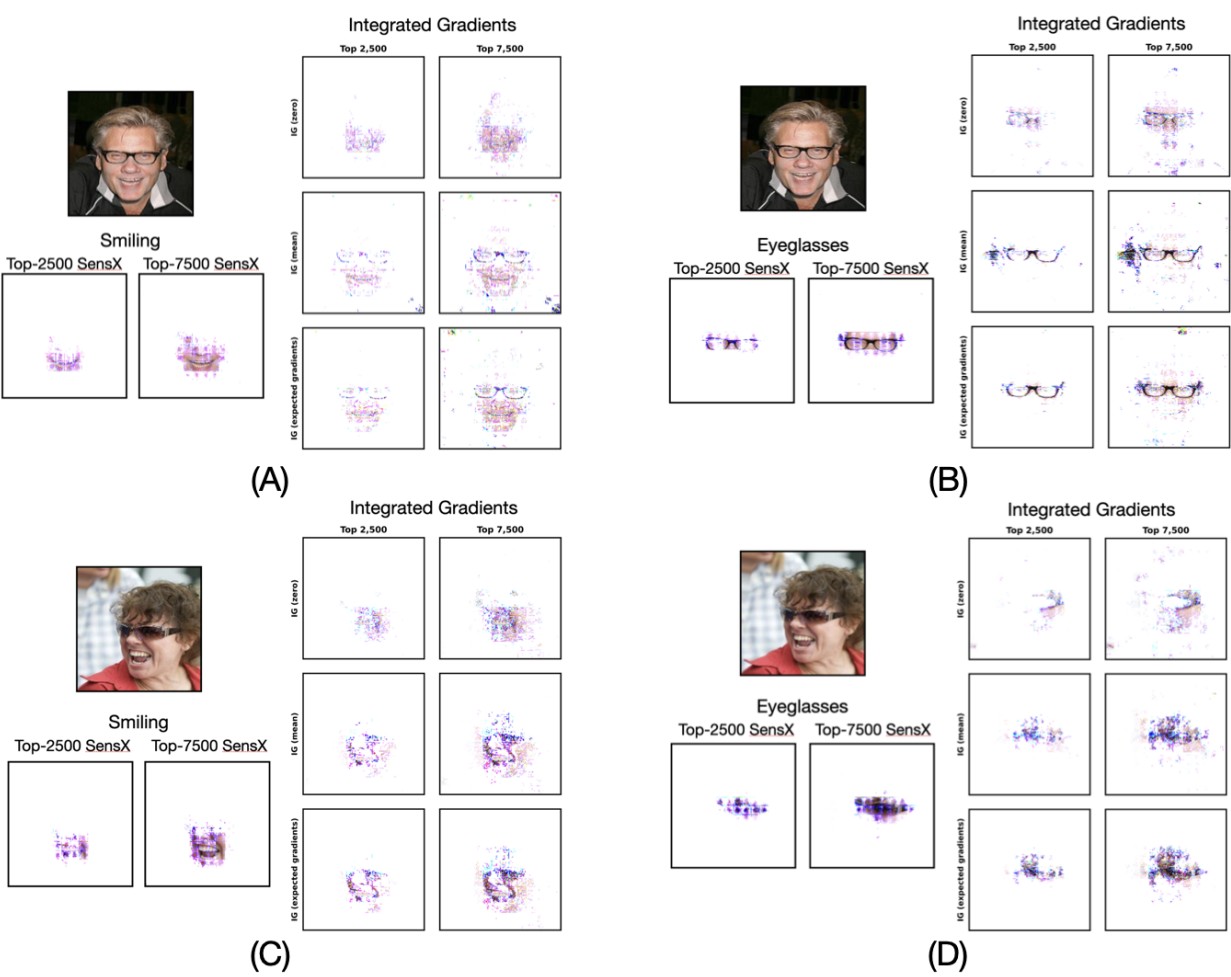

Figure D5: **SensX and IG top-$k$ masks for all (image, model) combinations.** Each quadrant shows one (image, model) combination with SensX masks (left) and IG masks under three baseline choices (right). **(A)** Image 000276, Smiling. **(B)** Image 000276, Eyeglasses. **(C)** Image 000375, Smiling. **(D)** Image 000375, Eyeglasses. SensX consistently localizes to task-relevant regions across all combinations. IG attributions vary with baseline choice and are less spatially coherent than SensX. On the smiling model (A, C), the mean baseline and expected gradients attribute to the eyeglass frames despite this model being fine-tuned independently from the eyeglasses model.

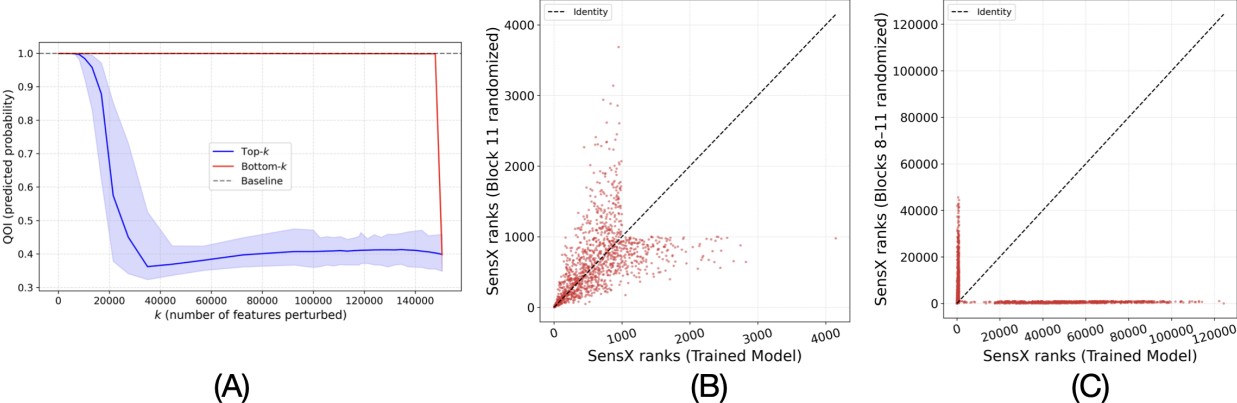

Figure D6: **Perturbation analysis and model randomization confirm SensX feature ranking.** All panels show results for the Eyeglasses model on image 000276. (A) Features are perturbed within the characteristic perturbation scale from highest to lowest (top-$k$, blue) and lowest to highest (bottom-$k$, red) SensX rank. Perturbing top-ranked features produces a rapid decline in QOI, while perturbing over 130,000 bottom-ranked features has no measurable effect. Shaded regions denote the 1st–99th percentile over 1,000 random perturbations. (B,C) Rank comparison between the trained model and models with progressively randomized weights. Each point represents a feature in the top 1,000 of either map. (B) Randomizing block 11 produces moderate rank disruption. (C) Randomizing blocks 8–11 destroys the ranking: top-ranked features in the trained model scatter to ranks above 20,000. Equivalent perturbation analyses for all four (image, model) combinations are shown in Figure D7.

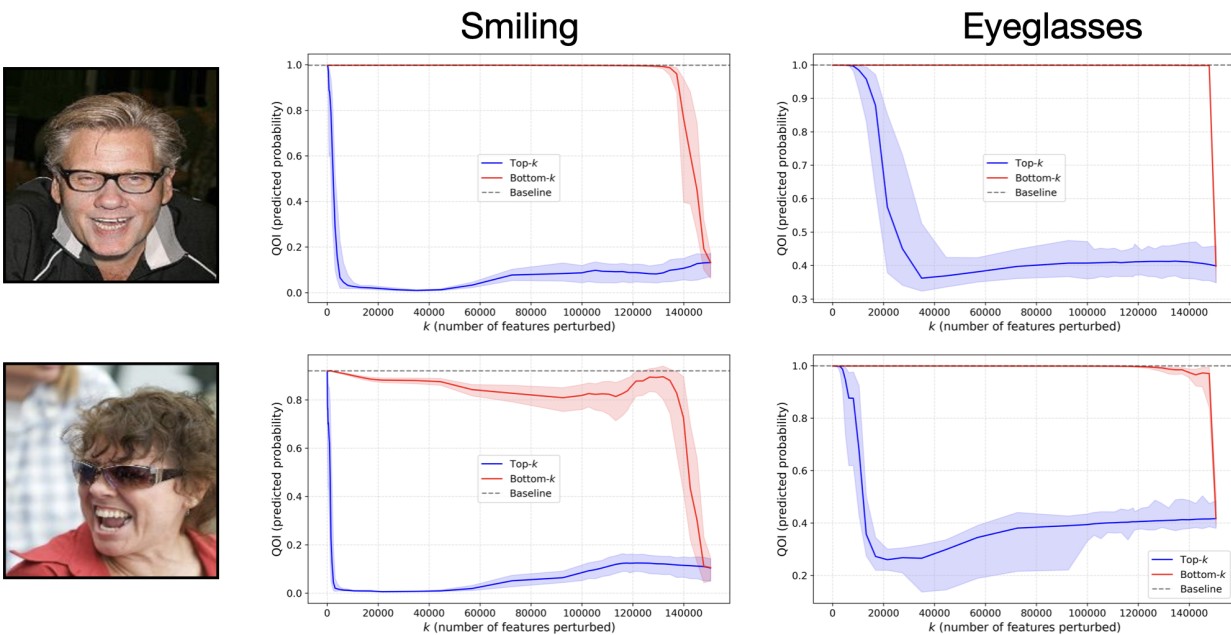

Figure D7: **Perturbation analysis confirms SensX feature ranking across all (image, model) combinations.** Each panel shows the QOI (predicted probability) as a function of the number of features perturbed within the characteristic perturbation scale, from highest to lowest (top-$k$, blue) and lowest to highest (bottom-$k$, red) SensX rank. Shaded regions denote the 1st–99th percentile over 1,000 random perturbations. Dashed line indicates the unperturbed baseline. Across all four cases, perturbing top-ranked features produces a rapid decline in QOI, while perturbing the majority of bottom-ranked features has no measurable effect. Both curves converge when all 150,528 features are perturbed.

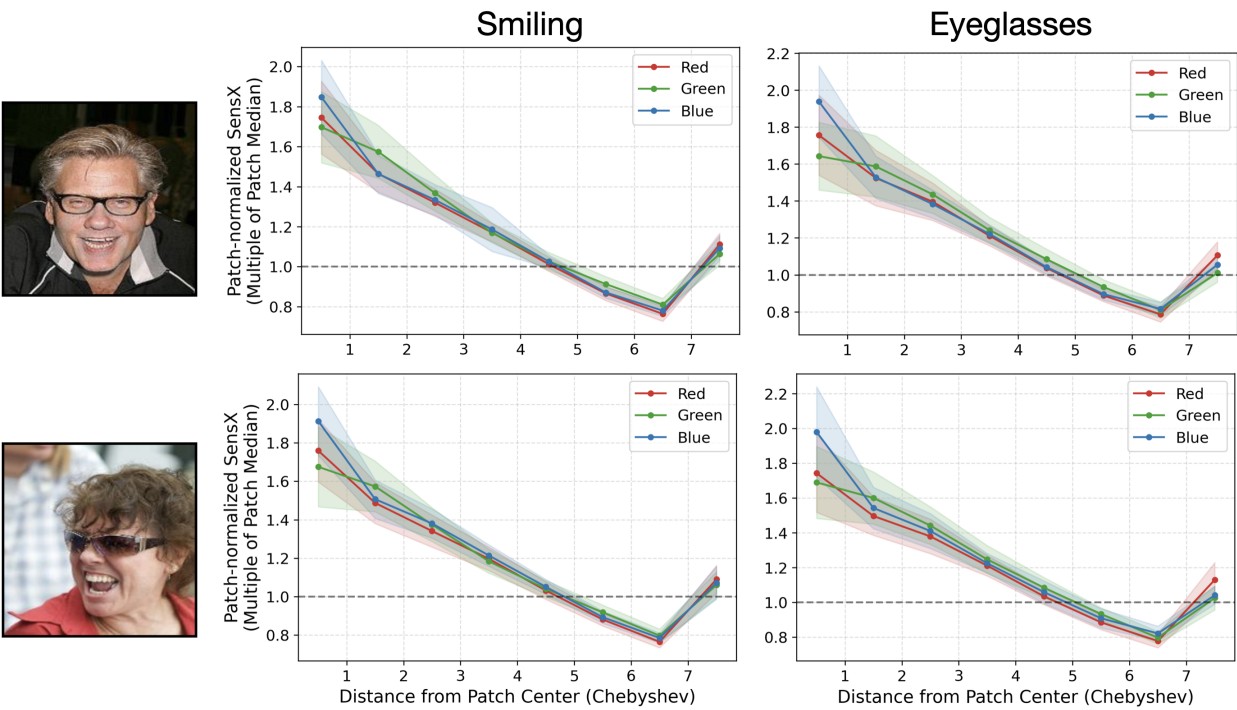

Figure D8: **Intra-patch spatial bias is consistent across images and models.** Patch-normalized SensX averaged across all 196 patches as a function of Chebyshev distance from the patch center, shown for both images (rows) and both models (columns). Shaded regions denote ±1 standard deviation. The profile shape (peak at center, decay to a minimum near distance 6, rebound at the patch boundary) is consistent across all four (image, model) combinations and all three color channels.

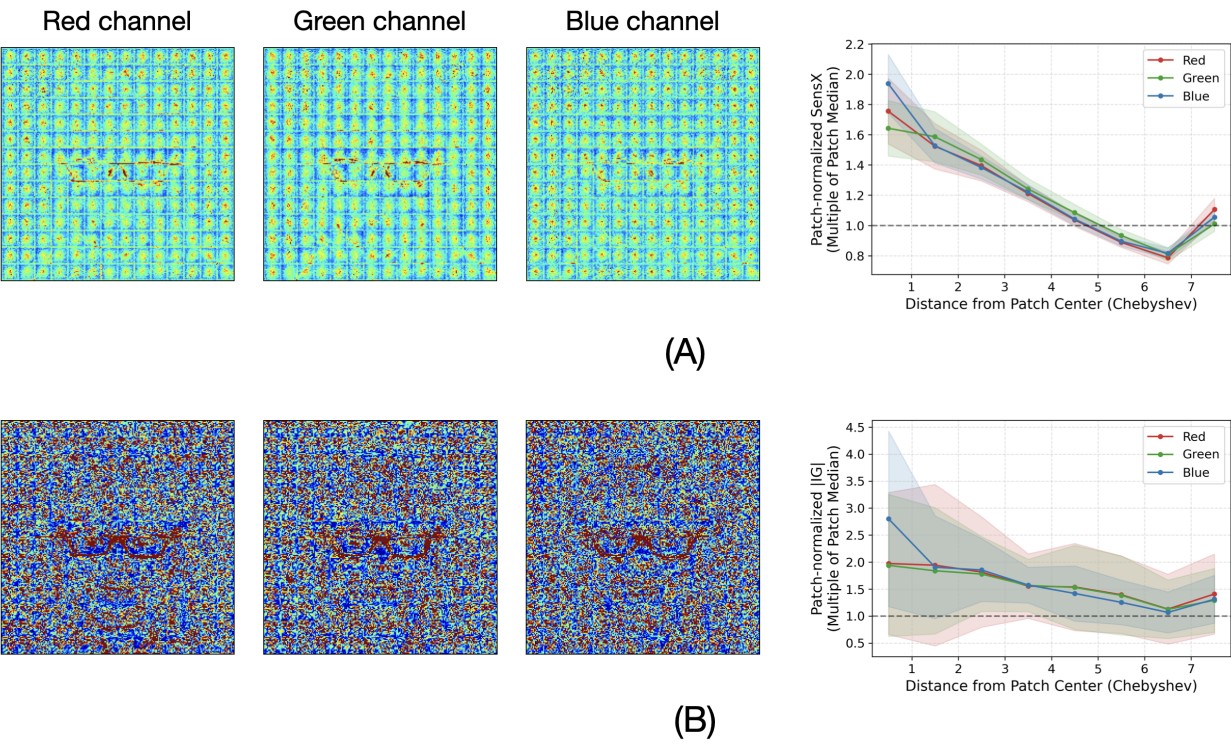

Figure D9: **IG does not resolve the intra-patch bias identified by SensX.** (A) SensX and (B) IG (expected gradients) for image 000276 under the Eyeglasses model. Left three columns: patch-median-normalized attribution heatmaps per color channel. Right column: patch-normalized attribution averaged across all 196 patches as a function of Chebyshev distance from the patch center, with shaded regions denoting $\pm 1$ standard deviation. SensX reveals a systematic intra-patch pattern (center peak, decay to a minimum near distance 6, rebound at the patch boundary) with narrow cross-patch variance. The IG heatmaps show no visible grid structure, and the IG distance profile exhibits a coarse center bias with substantially wider variance that obscures the non-monotonic pattern resolved by SensX.

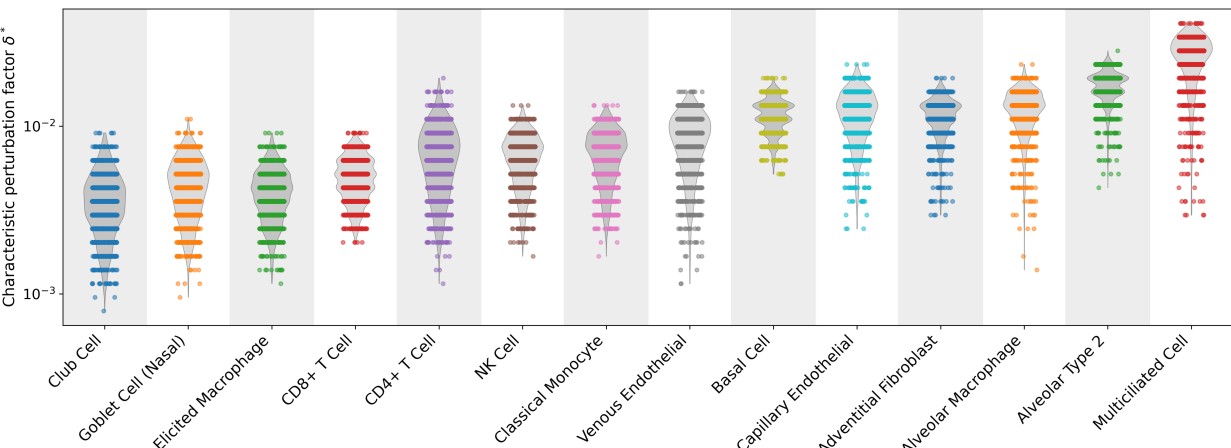

Figure D10: **Characteristic perturbation scale $\delta^*$ varies across and within cell types.** Distribution of $\delta^*$ for 1,000 high-confidence test cells (predicted probability $\geq 0.99$) per cell type, shown as violin plots with overlaid individual cells on a logarithmic scale. $\delta^*$ spans approximately two orders of magnitude ($10^{-3}$ to $10^{-1}$), with medians varying across cell types. The within-cell-type spread confirms that a fixed perturbation scale would be a poor approximation. SensX adapts $\delta^*$ per cell, capturing the local sensitivity structure learned by each classifier. Cell types are ordered by median $\delta^*$.

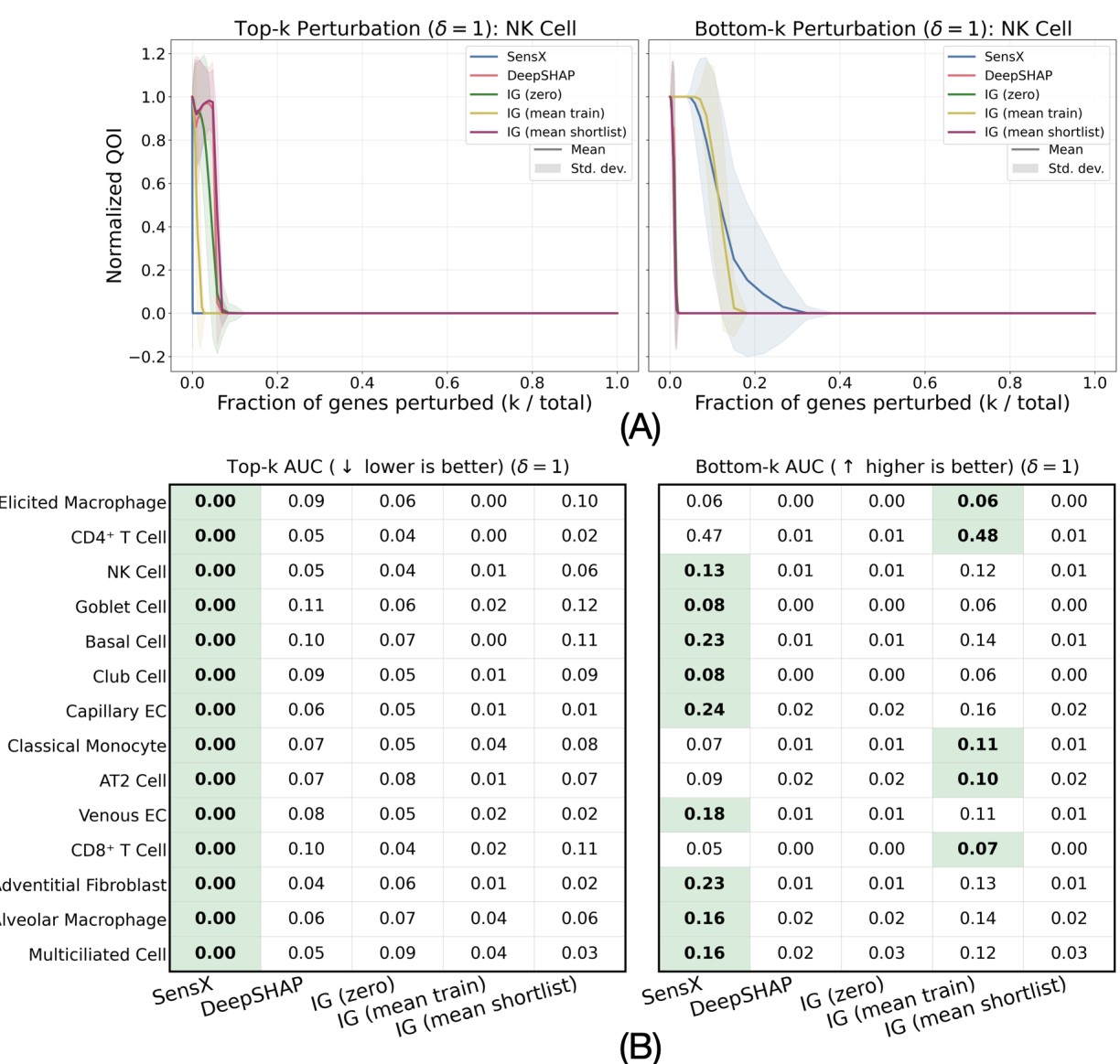

Figure D11: **Perturbation validation at $\delta = 1$ (global perturbation range).** (A) Top-$k$ (left) and bottom-$k$ (right) perturbation curves for natural killer cells, with each gene perturbed across its full global expression range. For each cell, the median predicted probability across 1,000 perturbations is recorded, normalized by the unperturbed baseline. Curves show the mean across 1,000 cells, with shaded regions denoting ± one standard deviation. (B) Normalized AUC of the perturbation curves for all 14 cell types. The best value per cell type is shown in bold. Cell types are sorted by SensX top-$k$ AUC. SensX top-$k$ AUC is 0.00 for all cell types, confirming that its gene rankings remain valid under global perturbation.

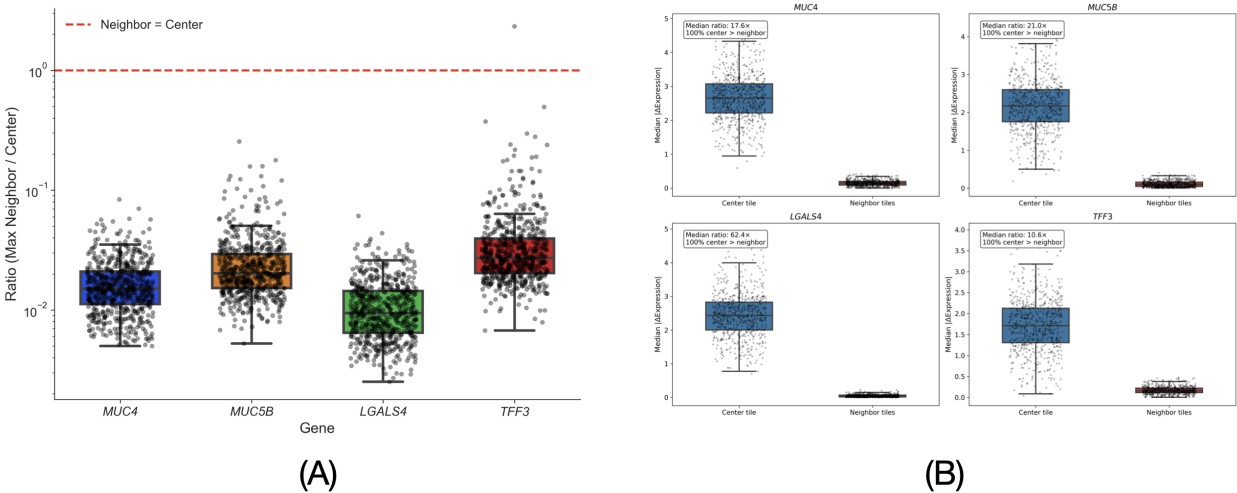

Figure D12: **Center-tile dominance in DeepSpot predictions.** (A) Ratio of the maximum neighboring-tile SensX to the center-tile SensX. The dashed red line indicates equal sensitivity. Nearly all ratios fall well below 1, with a median of 0.018. (B) Independent perturbation validation. For each spot-gene pair, the center tile or all eight neighbor tiles are perturbed within the characteristic perturbation scale (1,000 perturbations per condition), and the median absolute change in predicted log-expression is recorded. Perturbing the center tile produces substantially larger prediction changes than perturbing all neighbors across all four genes, with a median ratio of 20× and center-tile perturbation exceeding neighbor perturbation in 99.9% of spot-gene pairs.

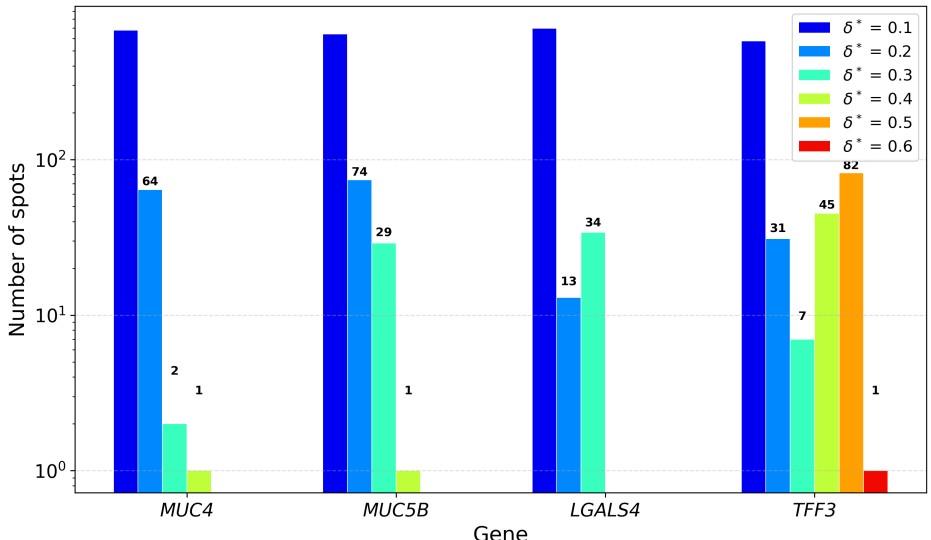

Figure D13: **Characteristic perturbation scale across spots in the histopathology case study.** Number of spots at each discrete $\delta^*$ value for four genes. The majority of spots have $\delta^* = 0.1$, the minimum value evaluated, across all genes. *TFF3* shows the broadest distribution, with 82 spots at $\delta^* = 0.5$.

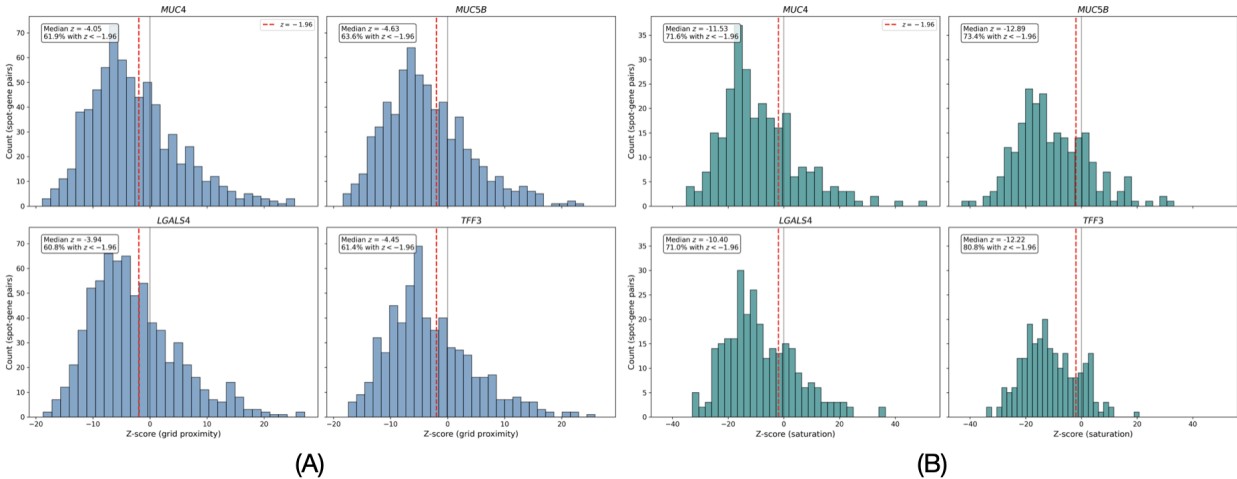

Figure D14: **Per-gene statistical confirmation of subspot grid bias and staining-saturation dependence.** (A) Distribution of $z$-scores from the subspot grid proximity test, shown separately for each gene. For each spot-gene pair, the mean distance from the top-500 SensX pixels to the nearest internal grid line is compared to a permutation null of 500 randomly selected pixels. Negative $z$-scores indicate that high-attribution pixels are significantly closer to grid boundaries than expected by chance ($z < -1.96$ corresponds to $p < 0.025$, one-tailed). (B) Distribution of $z$-scores from the saturation test, restricted to spot-gene pairs without significant grid bias. For each spot, the mean HSV saturation of the top-500 SensX pixels is compared to a permutation null of 500 randomly selected pixels. Negative $z$-scores indicate that high-attribution pixels have significantly lower saturation (lighter staining) than expected by chance. Dashed red line: $z = -1.96$ ($p = 0.025$). Both effects are consistent across all four genes.

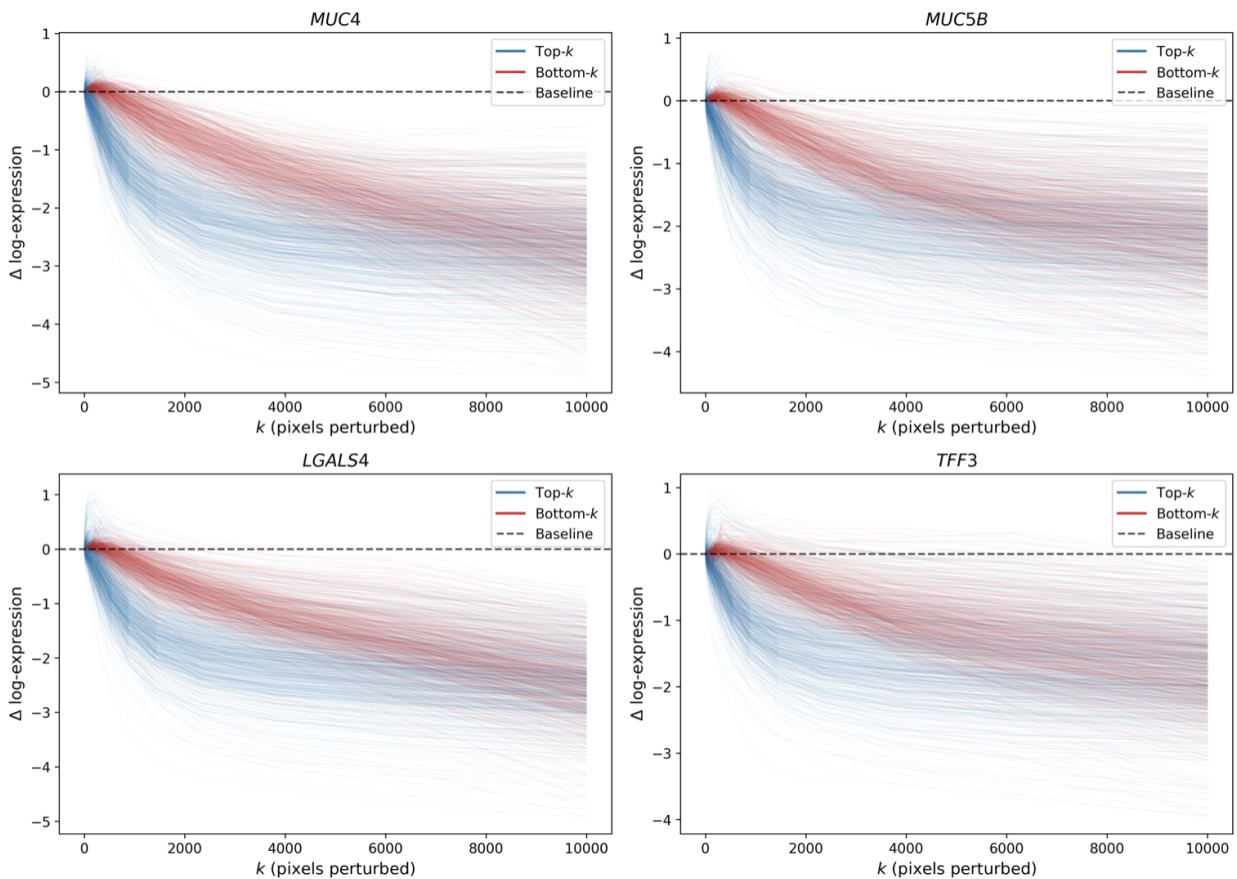

Figure D15: **Pixel-level perturbation validation of SensX ranking.** Change in predicted log-expression ($\Delta$) when perturbing the top-$k$ (blue) or bottom-$k$ (red) SensX-ranked pixels within the characteristic perturbation scale, for 20 log-spaced values of $k$ from 1 to 10,000 (100 perturbations per $k$). Each curve represents one spot. Perturbing a small number of top-ranked pixels produces large prediction changes, while perturbing thousands of bottom-ranked pixels has minimal effect, confirming that the pixel-level SensX ranking concentrates predictive importance into a small subset of pixels. Dashed black line: baseline (no perturbation).

# E    Ranking Convergence Analysis

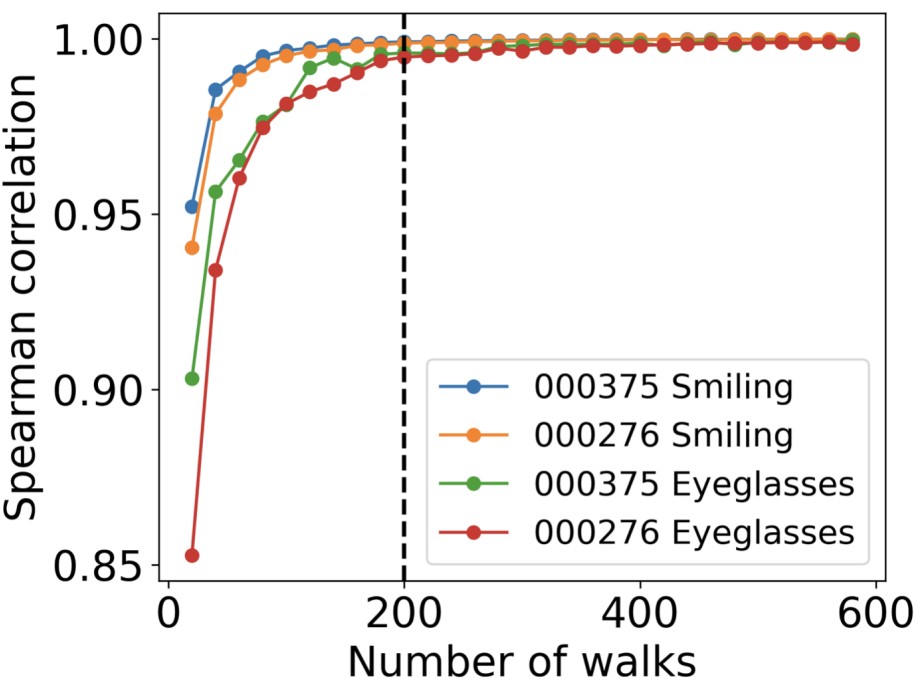

Figure E1: **SensX ranking convergence for the ViT case study.** We estimate SensX values cumulatively in batches of 20 coordinate-walks. At each cumulative walk count $n_w$, we compute the Spearman rank correlation between the feature rankings (higher SensX value gets a larger rank) obtained with $n_w$ walks and with $n_w - 20$ walks (i.e., after adding one additional batch). Curves are shown for the four (image, model) combinations. Rankings stabilize rapidly. Correlations exceed 0.99 by $\approx 200$ walks (dashed line), indicating diminishing returns from additional walks.

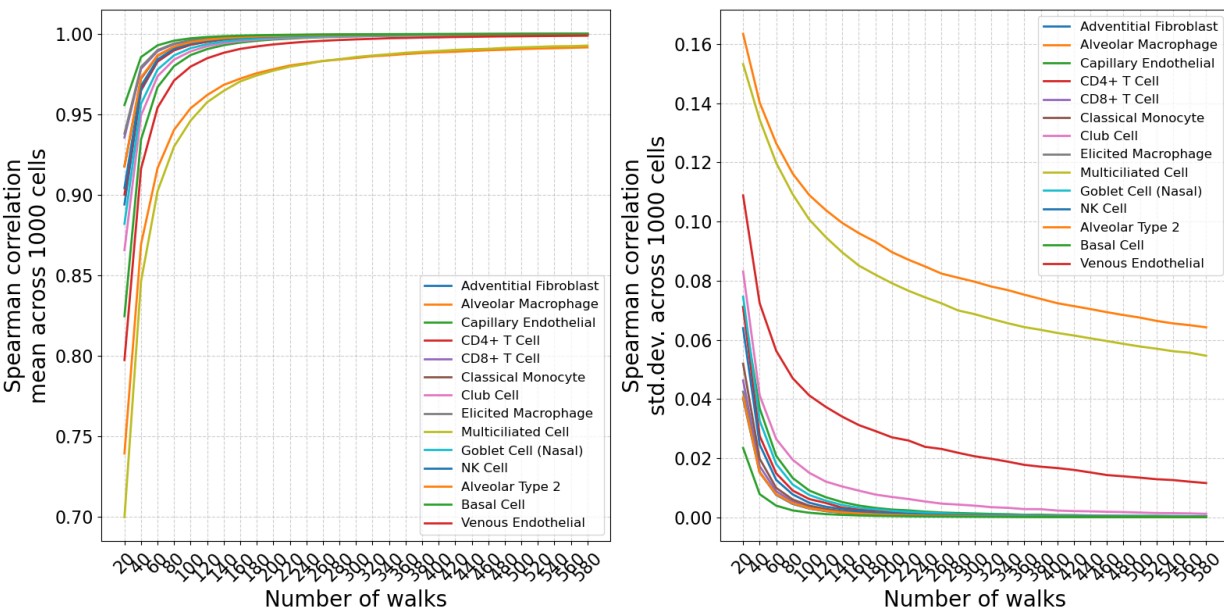

Figure E2: **Convergence of SensX gene rankings in single-cell classification.** SensX values are estimated cumulatively in batches of 20 coordinate-walks. At each cumulative walk count $n_w$, we compute the Spearman rank correlation between the gene rankings obtained with $n_w$ walks and with $n_w - 20$ walks (i.e., after adding one additional batch). Curves summarize 1,000 cells per cell type. (A) Mean correlation across cells. (B) Standard deviation across cells. Rankings stabilize rapidly for most cell types and continue to tighten with additional walks.

# F  Case Study Details

## F.1  Synthetic Datasets

Samples are generated from a ten-dimensional standard Gaussian distribution. Each sample, $\mathbf{x} = (x_1, \ldots, x_{10})$, is labeled $b \in \{0, 1\}$ based on the following criteria for three of the four synthetic datasets (Chen et al., 2018):

- XOR: $P(b = 1) \propto x_1 x_2$    ($k = 2$ important features)

- Orange skin: $P(b = 1) \propto \sum_{1 \leq i \leq 4} x_i^2 - 4$    ($k = 4$ important features)

- Nonlinear additive: $P(b = 1) \propto -100 \sin(0.2\, x_1) + |x_2| + x_3 + e^{-x_4} - 2.4$    ($k = 4$ important features)

The switch dataset is a mixture of data samples from two different generative processes. $x_{10}$ is generated from a mixture of two Gaussians centered at $\pm 3$ with equal probability. If $x_{10}$ is generated from the Gaussian centered at $+3$, then $x_1$ to $x_4$ are used to label the sample based on the orange skin model. Otherwise, $x_5$ to $x_8$ are used to label the sample based on the nonlinear additive model.

For all datasets, $N = 100{,}000$ samples were generated and split into 80% training and 20% test sets. The test set was further divided equally into validation and held-out test subsets.

**Model Architecture and Training.**  A feedforward neural network with three hidden layers (200, 100, and 50 units) and ReLU activations was trained independently on each dataset. The output layer produces a single logit, and binary cross-entropy with logits was used as the loss function. Models were trained for 50 epochs using the AdamW optimizer with a learning rate of $10^{-3}$, weight decay of $10^{-4}$, and a batch size of 64. The model with the highest validation F1 score was selected for downstream analysis. Feature attributions were computed on a subset of up to 1,000 test samples for which the trained model predicted class 1 with greater than 99% confidence.

**SensX Configuration.**  Sensitivity was computed using $n_w = n_s \in \{500, 1000, 2500, 5000, 7500, 10{,}000\}$ perturbation samples. The stability profile was evaluated over 20 equally spaced perturbation magnitudes $\delta \in [0.05, 1.0]$, and the characteristic $\delta$ was selected using a stability threshold of $\tau_a = 0.1$. Perturbation bounds were set to the feature-wise minimum and maximum across the full dataset.

**KernelSHAP Configuration.**  KernelSHAP was computed using a background dataset of $n_{\text{background}} = 100$ training samples summarized via $K$-means clustering, with coalition sample budgets of $n_{\text{samples}} \in \{500, 1000, 2500, 5000, 7500, 10{,}000\}$. Feature importance was taken as the absolute value of SHAP values.

**Integrated Gradients Configuration.**  Integrated gradients (IG) (Sundararajan et al., 2017) were computed using the Captum library (Kokhlikyan et al., 2020) with Gauss–Legendre quadrature and $n_{\text{steps}} \in \{50, 100, 200, 500\}$ integration steps. Three baseline choices were evaluated: the zero vector, the training-set mean, and a random training sample drawn independently per input. The zero and mean baselines are deterministic and were run once per configuration. The random baseline was repeated over 100 independent runs to match the evaluation protocol of SensX and KernelSHAP. Feature importance was taken as the absolute value of IG attributions.

**Evaluation.**  For SensX and KernelSHAP, each configuration was repeated over 100 independent runs to assess variability. Attribution accuracy was evaluated using a top-$k$ criterion. Prediction is scored as correct at rank $k$ only if the $k$ highest-ranked features all belong to the ground-truth set $\mathcal{S}$,

$$\text{Acc@}k = \frac{1}{N} \sum_{i=1}^{N} \mathbf{1}\left[ \left\{ \hat{f}_1^{(i)}, \ldots, \hat{f}_k^{(i)} \right\} \subseteq \mathcal{S} \right]. \tag{F1}$$

This metric is order-invariant. It requires the correct set of features to appear in the top $k$ but does not penalize their relative ranking. For the Switch dataset, accuracy was computed separately for each subpopulation

against its respective ground-truth set ($\mathcal{S}_{\text{orange}} = \{1, 2, 3, 4, 10\}$ and $\mathcal{S}_{\text{additive}} = \{5, 6, 7, 8, 10\}$), then combined across all samples. Bar plots report the mean accuracy at the best-performing hyperparameter for each method and convergence across hyperparameters is shown in Figures D1 and D2.

### F.2 Vision Transformer (ViT)

A pretrained ViT-Base model (Dosovitskiy et al., 2020) was downloaded from `google/vit-base-patch16-224-in21k` and fine-tuned to identify smiling and eyeglasses (two separate binary classifiers) using the Large-scale CelebFaces Attributes (CelebA) dataset (Liu et al., 2015) of 202,599 images. Details of model fine-tuning are available at the SensX code repository. Input images are $224 \times 224$ pixels across three color channels (RGB). Each feature is an individual color channel at a pixel location. The QOI is the predicted probability of belonging to the class.

**SensX Configuration.** Global perturbation bounds were set to $[0, 1]$, corresponding to the normalized pixel intensity range. The stability profile was evaluated over 50 equally spaced perturbation magnitudes $\delta \in [0.02, 1.0]$ with $n_s = 1,000$ samples. The characteristic $\delta$ was selected using a stability threshold of $\tau_a = 0.1$. Sensitivity was computed using $n_w = 600$ and batch-size 1000.

**Integrated Gradients Configuration.** IG was computed using the Captum library (Kokhlikyan et al., 2020) with Gauss–Legendre quadrature. Integration steps of $n_{\text{steps}} \in \{100, 200, 500\}$ were evaluated. The convergence delta (difference between the sum of attributions and the predicted output difference) was small at $n_{\text{steps}} = 500$, which was used for all reported results. Three baseline choices were evaluated. The zero baseline was set to the preprocessed zero image (accounting for ImageNet normalization). The mean baseline was computed as the pixel-wise mean of the training set after preprocessing. Expected gradients (Erion et al., 2021) averaged IG attributions over 100 random training images, each drawn independently as a per-input baseline. Feature importance was taken as the absolute value of IG attributions. The same top-$k$ mask visualization and intra-patch bias analysis applied to SensX were repeated identically for all three IG baselines.

**Perturbation Validation.** We performed perturbation validation (Appendix A.7) using 50 log-spaced values of $k$ from 1 to 150,528 with additional linear spacing between 100,000 and 150,528, and $N_p = 1,000$ perturbations per $k$.

**Model Randomization Check.** Following Adebayo et al. (2018), we progressively randomized ViT transformer blocks and recomputed SensX to verify that attributions depend on learned weights. Five cascading levels were tested: level 1 randomized block 11, level 2 randomized blocks 8–11, and levels 3–5 randomized progressively deeper blocks. For levels 3–5, the characteristic perturbation scale $\delta^*$ dropped to zero, indicating that the model output became insensitive to input perturbations. SensX was recomputed for levels 1 and 2 using the same configuration as the trained model.

**Intra-Patch Bias Analysis.** To quantify spatial bias introduced by the ViT's $16 \times 16$ patch tokenization, each patch in the attribution map was normalized by its median value. The normalized attribution was then computed as a function of Chebyshev distance from the patch center, averaged across all 196 patches per image. This analysis was applied to both SensX and all three IG baselines.

### F.3 Single-Cell Transcriptomics

**Dataset.** The core human lung cell atlas (HLCA) (Sikkema et al., 2023) was downloaded from the Human Cell Atlas data portal (`https://data.humancellatlas.org/hca-bio-networks/lung/atlases/lung-v1-0`), comprising $\sim 584$k cells. Cell type labels were taken from the `cell_type` field that maps HLCA cell populations to the standardized cell ontology terms and in some cases aggregates atlas-native subtypes. Genes with near-zero variance ($< 10^{-6}$) across all cells were removed, leaving 27,399 genes. Cell types with fewer than 10,000 cells were excluded, yielding 14 cell types for analysis.

**Model architecture and training.** For each cell type, a binary feedforward neural network with hidden layers $[250, 50]$, ReLU activations, and a single logit output was trained. The loss function was binary cross-entropy with logits, with class-specific positive weight equal to the ratio of negative to positive training samples. Models were trained for 10 epochs using AdamW (learning rate $10^{-4}$, weight decay $10^{-4}$) with a batch size of 64. The dataset was split into 70% training, 15% validation, and 15% test sets via stratified sampling. The model with the highest validation F1 score was selected. Details of model training are available at the SensX code repository.

**Global domain.** Per-gene minimum and maximum values were computed across the full dataset after applying the gene filter, defining the global domain $\Omega^g$ as the smallest hypercube containing the training data.

**High-confidence cell selection.** To determine which genes the trained models are using to predict cell types with high confidence, we sampled 1000 cells with predicted probability at least 0.99.

**SensX configuration.** The stability profile was evaluated over 50 log-spaced perturbation magnitudes $\delta \in [10^{-4}, 1]$ with $n_s = 1,000$ samples per $\delta$. The characteristic perturbation scale $\delta^*$ was selected using a stability threshold of $\tau_a = 0.1$. SensX was computed with $n_w = 600$ and batch-size 8192.

**DeepSHAP configuration.** DeepSHAP (Lundberg and Lee, 2017) was computed using a background dataset of 100 samples drawn from the high-confidence cells of each cell type. SHAP values were computed in batches of 32 with additivity verification enabled. For each cell, genes were ranked by the absolute SHAP values.

**Integrated Gradients configuration.** Integrated Gradients (IG) (Sundararajan et al., 2017) was computed using Captum with $n_{\text{steps}} = 200$ and Gauss-Legendre quadrature. Three baseline variants were tested: (i) zero vector, (ii) mean of the training data, and (iii) mean of the high-confidence cells. For deterministic baselines (i–iii), a single baseline was shared across all cells. For each cell, genes were ranked by the absolute IG attribution. Convergence was verified via the completeness error returned by Captum.

**Perturbation validation.** We performed perturbation validation (Appendix A.7) for each method independently. For each cell, genes were ranked by SensX value, absolute DeepSHAP value, or absolute IG attribution. The top-$k$ and bottom-$k$ perturbation analyses were applied to each cell independently. We used 55 linearly spaced values of $k$ from 1 to 27,399 and $N_p = 1,000$ perturbations per $k$. For each cell, all methods were evaluated under the same perturbation protocol and the cell's $\delta^*$. All methods attribute the same quantity of interest (predicted probability via sigmoid).

### F.4  Spatial Transcriptomics (DeepSpot)

DeepSpot (Nonchev et al., 2025) predicts spatially resolved gene expression from H&E histology images. For each spot, defined as a $100 \times 100$ pixel tile, the model integrates visual information from the center tile and its eight neighboring tiles. The center tile is additionally partitioned into a $3 \times 3$ grid of subspots. The UNI pathology foundation model (Chen et al., 2024) extracts morphological features from all 18 regions (9 tiles + 9 subspots), which secondary architectures integrate to predict gene expression at the center tile. A pretrained DeepSpot model for colon tissue (Colon_HEST1K) was applied to a whole-slide H&E image (sample ZEN38) from the DeepSpot repository, yielding 745 spots with eight neighbors each.

**Gene Selection.** Four genes were selected for analysis based on high mean and high variance of predicted log-expression across spots: *MUC4*, *MUC5B*, *LGALS4*, and *TFF3*. Each QOI is the predicted log-expression of a single gene.

**Tile-Level SensX.** The RGB values of the center and neighboring tile pixels total to 270,000 features ($(1 + 8) \times 100 \times 100 \times 3$). Subspots are derived from the perturbed center tile. The entire DeepSpot pipeline, including UNI foundation model feature extraction, is treated as a black box from raw pixels to predicted

expression. The 270,000 features were grouped into 9 tiles of 30,000 features each, yielding one SensX value per tile. Global perturbation bounds were set to $[0, 255]$. Sensitivity was computed using $n_w = 500$ walks with a batch size of 128.

**Center-Tile Perturbation Validation.** To independently validate center-tile dominance, we perturbed either the center tile (30,000 features) or all eight neighbor tiles (240,000 features) uniformly within the local perturbation bounds, holding the other fixed. For each condition, $N_p = 1,000$ perturbations were drawn and the median absolute change in predicted log-expression was compared across conditions for each spot-gene pair.

**Pixel-Level SensX.** Following the tile-level finding that the center tile dominates, pixel-level analysis was restricted to the center tile with neighboring tiles held fixed. The three color channels at each spatial coordinate were grouped, yielding 10,000 feature groups per spot. SensX was computed using $n_w = 300$ walks.

**Pixel-Level Perturbation Validation.** We performed perturbation validation (Appendix A.7) on the 10,000 pixels within each center tile using 20 log-spaced values of $k$ and $N_p = 100$ perturbations per $k$. Because the SensX ranking differs across genes, perturbations were performed independently for each gene. Tasks sharing the same $\delta^*$ at a given spot were grouped to reduce redundant model evaluations.

**Subspot Grid Bias Test.** To test whether high-attribution pixels concentrate near the internal boundaries of DeepSpot's $3 \times 3$ subspot grid, we computed the minimum distance from each pixel to the nearest internal grid line (at pixel coordinates $33.\overline{3}$ and $66.\overline{6}$ in each dimension). For each spot-gene pair, we compared the mean grid-line distance of the top-500 SensX pixels to a null distribution of mean distances from 500 pixels sampled uniformly at random (10,000 permutations). The null distribution is independent of spot or gene identity and was computed once. Significance was assessed via a one-tailed permutation test at $p < 0.025$.

**Staining Saturation Bias Test.** To test whether high-attribution pixels are biased toward regions of lower staining intensity, we restricted the analysis to spot-gene pairs without significant grid bias ($p \geq 0.025$ in the grid proximity test). For each spot, the center tile was converted from RGB to HSV color space, and the saturation channel was extracted. We compared the mean saturation of the top-500 SensX pixels to a null distribution of mean saturations from 500 pixels sampled uniformly at random (10,000 permutations per spot). The null was computed once per spot and shared across genes, as saturation depends only on the tile image. Significance was assessed via a one-tailed permutation test at $p < 0.025$.

