# OpenReview forum: "SensX: Model-Agnostic Local Feature Attribution via Calibrated Global Sensitivity Analysis"
_TMLR — Accepted by TMLR_

### Review · Reviewer_BbrW · 2026-03-15

**Summary Of Contributions:**

This paper proposes SensX, a local feature-attribution method for black-box predictors that only requires forward evaluations and admissible feature bounds. The method defines a local hyper-rectangular neighborhood, chooses an input-specific perturbation scale via a saturation criterion on the prediction envelope, and then computes anchored Morris-style mean absolute elementary effects over grounded coordinate-wise trajectories starting at the input of interest. The paper also includes a grouped variant and a perturbation-based validation protocol. The model is validated empirically.

**Additional Comments:**

I am positive on the paper, but for theory-oriented readers it is currently "a precise method plus strong experiments,'' not yet "a theoretically grounded attribution framework.'' If the authors can either add modest formal results or temper the stronger theory-adjacent claims, I would be comfortable endorsing publication.

**Audience:**

Yes

**Audience Explanation:**

There is a clear audience in interpretability, trustworthy ML, computational biology, and practitioners working with chained, frozen, or API-only systems. The paper makes a persuasive case that existing attribution methods break down when the computational graph is unavailable or when no defensible reference distribution exists. It also makes a broader scientific point that attribution resolution matters: the ViT intra-patch artifact and the spatial-transcriptomics grid effect only become visible at per-feature resolution.

**Claims And Evidence:**

Yes

**Claims Explanation:**

The empirical claims are supported well. The paper is precise about the method, benchmarks against reasonable baselines where applicable, uses perturbation validation rather than relying purely on visual inspection, and includes an Adebayo-style cascading-randomization sanity check in the ViT case. For the claims the paper actually demonstrates experimentally, the evidence is convincing.

My caveat concerns the paper's more theory-adjacent language. Terms such as ``calibrated,'' ``principled,'' and ``interpretable'' are justified operationally more than mathematically. Equation (2) gives a well-defined rule for choosing \(\delta_x^\ast\), but there is no theorem showing consistency, sample complexity, stability under refinement of \(\Delta\), or conditions under which the resulting score recovers a meaningful target. Likewise, the anchored \(\mu^\ast\) construction is reasonable, but the paper does not formally characterize whether SensX approximates an average absolute partial derivative, a local ANOVA-type quantity, or some other canonical local functional. Thus, I believe the empirical claims are well supported, but the theoretical interpretation should either be strengthened or stated more modestly.

**Requested Changes:**

**Characterize the estimand more formally.** The most important missing piece is a theorem-level interpretation of SensX. For example, under smoothness assumptions, can the authors show that as \(\delta_x^\ast \to 0\) and \(n_w \to \infty\), SensX converges to a local quantity related to \(|\partial_j q(x)|\) or to an expectation of \(|\partial_j q(z)|\) over an induced local distribution? Exact or asymptotic results for linear, additive, or separable models would be a nice addition.

**Provide some finite-sample or asymptotic analysis for the calibration step.** The rule defining \(\delta_x^\ast\) is interesting, but the paper gives no statistical analysis of the stability profile based on \(n_s\) Monte Carlo samples. Even a modest result (for example, concentration of the estimated envelope, monotonicity properties under regularity assumptions, or stability under grid refinement) would significantly strengthen the paper.

**Discuss which attribution axioms or properties SensX satisfies or cannot satisfy.** SensX uses mean absolute elementary effects and therefore deliberately discards sign. That is a legitimate design choice, but it has consequences: completeness is absent, directional effects are lost, and interacting features may both receive large mass. The paper should explicitly discuss which classical desiderata are relevant here and where SensX stands with respect to them.

**Be more explicit about dependence on \(\Omega_g\), \(\Delta\), and \(\tau_a\).** The paper argues that these are interpretable knobs, and I agree they are more transparent than arbitrary baselines. But they still induce the perturbation regime and therefore materially affect the result. A clearer analysis of sensitivity to these choices, including edge cases such as \(\delta_x^\ast = 0\), sharp non-smooth response surfaces, and discrete or constrained inputs where independent perturbation is not straightforward, would be a nice addition.


*The following is optional, but would make the paper even stronger*

**State computational cost more plainly.** The memory comparison to KernelSHAP is persuasive, but the runtime burden should be made equally visible. In the ViT case, the reported 5 minutes per walk with \(n_w=600\) corresponds to a substantial audit cost per image-model pair on a single A100. This does not invalidate the method, but readers should see the cost-performance tradeoff as clearly as they see the memory story.

---

> ### Author Response · Authors · 2026-04-03
> **Response to Requested changes 1 and 2**
>
> **Requested change 1: Characterize the estimand more formally.**
>
> We have added three formal results (Propositions 1–3, Appendix A).
>
> - **Proposition 1 (linear models).** For $q(\mathbf{x})=\mathbf{w}^\top\mathbf{x}+b$, the elementary effect of feature $j$ equals $w_j$ along every grounded trajectory for any $\delta^\ast$ and any $n_w\geq1$, so $\hat{s}_{\mathbf{x},j}=|w_j|$ exactly.
> - **Proposition 2 (nonlinear additive models).** For $q(\mathbf{x})=\sum_{j} g_j(x_j)$, the elementary effect of feature $j$ depends only on the endpoint $e_j$ and not on permutation order or the values of other features. As $n_w\rightarrow\infty$ the SensX value converges almost surely to $\mathbb{E}_{e_j}[|g_j(e_j)-g_j(x_j)|/|e_j-x_j|]$. As $\delta^\ast\rightarrow0$ additionally, the sampling interval for $e_j$ contracts to $\{x_j\}$ and this expectation converges to $|g_j'(x_j)|$.
> - **Proposition 3 (general continuously differentiable models).** As $\delta^\ast\rightarrow0$, all endpoints collapse to $\mathbf{x}$, making permutation order irrelevant and causing each elementary effect to converge uniformly to $\partial_j q(\mathbf{x})$ over all trajectories, so $\hat{s}_{\mathbf{x},j}\rightarrow|\partial_j q(\mathbf{x})|$ for any fixed $n_w\geq1$.
>
> For nonlinear models with interactions, $\delta^\ast$ may be bounded away from zero. Propositions 1 and 2 are valid in this regime for linear and additive models, but no analytic form for the estimand exists for general non-additive models. We state this explicitly as an open problem in Section 3.3 and Appendix A.
>
> **Requested change 2: Provide some finite-sample or asymptotic analysis for the calibration step.**
>
> We have added a formal stability analysis in Appendix A (Finite-Sample Stability of the Characteristic Perturbation Scale), addressing each of the three directions suggested.
>
> - **Monotonicity.** The saturation criterion that defines $\delta^\ast$ asks whether the widest prediction envelope observed at any scale above a candidate $\delta'$ is below $\tau_a$. Because increasing $\delta'$ shrinks the set of scales inspected, the envelope width can only decrease or stay the same. We formalise this as Proposition 5: the saturation criterion is monotonically non-increasing in $\delta'$.
> - **Concentration.** At each grid point, the 1st and 99th percentiles of the QOI distribution are estimated from $n_s$ i.i.d. samples. The Dvoretzky–Kiefer–Wolfowitz inequality controls empirical CDF accuracy. Translating this into quantile accuracy requires that the QOI distribution has positive density at the relevant percentiles — a condition satisfied whenever $q$ is continuous and the perturbation distribution is absolutely continuous. After this density correction and a union bound over all grid points, the estimated $\delta^\ast$ recovers the population $\delta^\ast$ with an explicit sufficient sample size that scales as $O(\log G)$, where $G$ is the number of grid points, with the constant depending on the minimum QOI density at the percentiles and on how closely the saturation criterion approaches $\tau_a$ near the transition.
> - **Grid refinement.** When the true transition falls between grid points, we provide a bound on the resulting discretization error (Proposition 5): if the saturation criterion is strictly decreasing near the true transition, then the discrete estimate overshoots by at most the local grid spacing at the transition. The strict monotonicity condition can fail if the saturation criterion plateaus (i.e., the model saturates abruptly over a range of scales rather than at a single point). In such cases the bound does not apply and the discretization error depends on the length of the plateau.
>
> Appendix B (Perturbation Factor Grid Selection) demonstrates this concretely for the single-cell classifiers, where the sensitivity transition occurs near zero: a uniform grid returns $\delta^\ast = 0$ because all grid points fall in the saturated plateau, while a log-spaced grid concentrates resolution near the transition, producing small local spacing where it matters, and recovers a stable non-zero $\delta^\ast$ across all grid resolutions and sample sizes. At the default $n_s = 1{,}000$, the dominant source of error is grid placement rather than sample size.

---

> ### Author Response · Authors · 2026-04-03
> **Response to Requested changes 3, 4, and 5**
>
> **Requested change 3: Discuss which attribution axioms or properties SensX satisfies or cannot satisfy.**
>
> We have added a paragraph on attribution axioms to the Discussion (Relationship to attribution axioms) and expanded the Limitations paragraph (Axiomatic).
>
> SensX satisfies sensitivity, implementation invariance, symmetry in expectation, and the dummy property. It does not satisfy completeness and provides no directional information — both consequences of using mean absolute elementary effects. The intended use is ranking-oriented: high-$\mu^*_j$ features provide a reduced subset for downstream sign-sensitive analyses. Details are in the Relationship to attribution axioms paragraph in the Discussion and the Axiomatic section of Limitations.
>
> **Requested change 4: Be more explicit about dependence on $\Omega_g$, $\Delta$, and $\tau_a$.**
>
> The Discussion now includes a dedicated analysis of each parameter's edge cases and failure modes (Hyperparameter interpretability paragraph). Appendix B analyzes the dependence of calibration on $n_s$, $\Delta$, and $\tau_a$ for the single-cell case study and shows the edge case of $\delta_\mathbf{x}^\ast=0$. Requirement of $\Omega^g$ as a reference is now stated explicitly in the comparison of requirements paragraph of the Related Work section.
>
> - **$\tau_a$:** encodes a domain-meaningful tolerance for output variation. In the limit of infinite $n_s$ and infinitely fine $\Delta$ for a smooth model, $\delta_\mathbf{x}^\ast$ is non-decreasing as $\tau_a$ decreases — a stricter threshold requires the model to have saturated more tightly. In practice, finite grid resolution and sampling noise mean the effect of changing $\tau_a$ on $\delta_\mathbf{x}^\ast$ depends on the model's sensitivity profile and $\Delta$. The edge cases are: when too strict, $\delta_\mathbf{x}^\ast=0$ for inputs in flat prediction regions, an explicit diagnostic rather than a silent failure; when too loose, $\delta_\mathbf{x}^\ast\to 1$ and perturbations span the full domain $\Omega^g$.
> - **$\Delta$:** a uniform grid sufficed in most settings; a log-spaced grid was used for single-cell where $\delta_\mathbf{x}^\ast$ clustered near zero (Appendix B). Sharp non-smooth response surfaces pose a challenge for any finite grid: a sensitivity transition that falls between grid points will cause $\delta_\mathbf{x}^\ast$ to be sensitive to grid placement, and the practical diagnostic is stability of $\delta_\mathbf{x}^\ast$ under grid refinement.
> - **$\Omega^g$:** setting it too tightly risks missing genuine sensitivity outside the bounds; setting it too broadly includes implausible feature values.
>
> Discrete inputs are a separate open problem: the elementary effect requires a nonzero denominator and a continuous perturbation path, neither of which extends naturally to binary, ordinal, or categorical features. We acknowledge this in the practical limitations paragraph in the Discussion. The question of constrained inputs, where independent perturbation moves the input off a constraint surface or data manifold, is addressed by the manifold objection paragraph in the Discussion.
>
> **Requested change 5: The following is optional, but would make the paper even stronger: State computational cost more plainly.**
>
> Table 2 now reports per-walk time and total cost per input for the real data case studies on a single A100 GPU. Per-walk time varies by orders of magnitude across case studies (~5 min per walk for the ViT and 20 walks per second for single-cell), driven by batched forward pass cost rather than feature count. Total cost scales linearly with the number of walks, which is determined by the ranking convergence analysis or the perturbation validation protocol rather than a fixed budget.

---

> > ### Comment · Action_Editor_SaGp · 2026-04-13
> >
> > Dear Reviewer,
> >
> > Could you please have a look at the authors' response and update your review accordingly ?
> >
> > Best
> > Amartya

---

### Review · Reviewer_19fL · 2026-03-29

**Summary Of Contributions:**

Summary
This paper proposes a novel method for local feature attribution tailored to black-box and composite models. The approach is designed to operate without requiring access to model internals, gradients, or training data.

Contributions

1. The method relies solely on forward evaluations, avoiding the need for gradients, internal model access, or training data.
2. It introduces an adaptive perturbation scale parameter to improve attribution quality.

Strengths

1. The paper includes detailed experiments with clear and well-structured descriptions.
2. The method is broadly applicable to black-box systems, making it practical in real-world scenarios.

Weaknesses

1. Theoretical guarantees are limited compared to methods such as Kernel SHAP, which may raise concerns about interpretability rigor.

**Audience:**

Yes

**Audience Explanation:**

Given the increasing need for model-agnostic interpretability in complex systems, I believe this work will be of interest to a broad TMLR audience.

**Broader Impact Concerns:**

I did not find any concern.

**Claims And Evidence:**

Yes

**Claims Explanation:**

The paper presents clear empirical results across both synthetic benchmarks and real-world applications, showing that the proposed method works well in a range of settings. The experiments are carefully designed and generally support the main claims. I believe this work is supported by accurate, convincing and clear evidence.

**Requested Changes:**

1. A brief discussion of limitations would help provide a more balanced perspective. For example, it would be useful to understand in which settings SensX may be less effective, such as models with highly non-smooth behavior or strong feature dependencies.
2. It would also be helpful to clarify the details of the permutation-based testing procedure. Eg, in page 8, before mentioning "p < 0.025", we need to state the null and alternative hypothesis,

---

> ### Author Response · Authors · 2026-04-03
>
> **Requested change 1: Limitations discussion.**
>
> We have added a dedicated Limitations paragraph to the Discussion, organized into three subsections, briefly outlined below.
>
> *Theoretical.* Propositions 1–3 address the $\delta^\ast \to 0$ regime (exact recovery for linear models, a.s. convergence for additive models, uniform convergence to $|\partial_j q(x)|$ for $C^1$ models). The open problem for general non-additive models at finite $\delta^\ast$ is stated explicitly in Section 3.3 and Appendix A. No concentration bound on ranking convergence is established.
>
> *Axiomatic.* SensX does not satisfy completeness and does not provide directional information. The intended use is ranking-oriented. The high-$\mu_j^\ast$ features identified provide a reduced subset on which sign-sensitive or directional analyses can subsequently be applied. Features involved in strong interactions can both receive large values.
>
> *Practical.* Computational feasibility depends on the cost of batched forward model evaluations. The coordinate-wise walk structure does not extend naturally to discrete inputs such as binary, ordinal, or categorical features, and we leave this to future work. Sharp non-smooth response surfaces can cause $\delta^\ast$ to be sensitive to grid placement, with a log-spaced grid or refinement sweep as the practical remedy (updated Appendix B shows a concrete example). The global $\Omega^g$ domain requires domain knowledge to specify.
>
> **Requested change 2: Null and alternative hypotheses for the permutation tests.**
>
> We have added explicit null and alternative hypotheses for both tests in the DeepSpot results section, before the $p$-values.
>
> *Grid proximity test.* Null: high-attribution pixels are distributed uniformly over the center tile. Alternative (one-sided): high-attribution pixels concentrate closer to the internal grid lines than expected under the null. One-tailed permutation test; null distribution of mean grid-line distances from 500 uniformly sampled pixels, 10,000 permutations, computed once and shared across all spot–gene pairs.
>
> *Staining saturation test.* Null: high-attribution pixels are distributed uniformly over the center tile. Alternative (one-sided): high-attribution pixels have lower staining saturation than expected under the null. One-tailed permutation test; null distribution of mean saturation from 500 uniformly sampled pixels, 10,000 permutations per spot, shared across genes.
>
> **On the weakness regarding theoretical guarantees:**
>
> We have added three formal results (Propositions 1–3, Appendix A). SensX provides exact recovery for linear models (Proposition 1), almost-sure convergence for additive models (Proposition 2), and convergence to $|\partial_j q(x)|$ for any continuously differentiable model in the $\delta^\ast\rightarrow 0$ limit (Proposition 3).
>
> When $\delta^\ast$ is bounded away from zero with general non-additive models, no analytic form for the estimand exists, and we state this explicitly as an open problem in Section 3.3 and Appendix A.
>
> We note that KernelSHAP's axiomatic guarantees are conditional on a fixed background dataset and hold exactly only in the limit of infinite coalition samples. In practice both the background choice and the finite sample budget materially affect the attributions.

---

> > ### Comment · Reviewer_19fL · 2026-04-03
> > **Thank authors for their response**
> >
> > The discussion looks good. Thanks for the explanations.

---

### Review · Reviewer_nG1P · 2026-03-30

**Summary Of Contributions:**

__Paper Summary__

This paper introduces a new feature attribution method, SensX, that first identifies an input-specific perturbation set and then adapts a Morris-style coordinate-walk strategy for local attribution. SensX addresses two limitations of prior methods: (1) unlike gradient-based methods (e.g., IG), it does not require model internals, such as the end-to-end computational graph for calculating gradients, and (2) compared with other perturbation-based methods (e.g., KernelSHAP), it offers a more system-grounded perturbation scheme. Experimental results demonstrate the effectiveness of SensX, outperforming baselines such as KernelSHAP and IG by a large margin, while validating its generality across black-box, composite systems.

__Strengths__

+ The paper is well motivated, building on legitimate, clearly stated critiques of prior feature-attribution methods. Developing a reliable feature-attribution method applicable to general black-box, composite AI systems is highly relevant in practice. In this regard, the paper is timely and focuses on an important research question.

+ The idea of replacing arbitrary design choices with more interpretable, application-grounded approaches to design the perturbation distribution for each local input is sensible. The adaptation of Morris-style coordinate perturbations to local attribution is intuitive and looks novel.

+ The paper presents a range of thoughtful case studies (synthetic, vision transformers, single-cell data, and composite pipelines) that provide strong empirical support for the main arguments in terms of generality. Compared with baselines, SensX clearly outperforms them in terms of attribution accuracy and reliability.

__Weaknesses__

- The biggest weakness of the paper, in my opinion, is the incomplete coverage of related methods. The paper focused on baseline methods that are quite old (IG, KernelSHAP, DeepSHAP), which are inadequate given the field’s recent advancements (see [1,2] for a survey of different Shapley-based attribution methods). Consequently, the situation of SensX within the broader literature is vague. The paper can be significantly strengthened if it includes a related work section that systematizes recent advancements / clarifying the paper’s positioning, and compares SensX’s performance with more state-of-the-art feature attribution methods.

- The general critiques of existing methods are valid, but are rhetorically overstated in the paper. While IG does require gradient access, it remains applicable to submodules via differentiable surrogate modeling in practical settings (see [3]). While KernelSHAP depends on a background distribution, there exist established heuristics (e.g., dataset-based baselines [4] and conditional sampling [5]) that often yield useful results. The claim that these methods are not applicable in composite or modular systems is too strong, as it may not reflect the current practice (see [6]). I recommend that the paper use more cautious language to avoid overassertive statements.


[1] Algorithms to estimate Shapley value feature attributions, https://arxiv.org/pdf/2207.07605

[2] A comparative study of methods for estimating model-agnostic Shapley value explanations, https://link.springer.com/article/10.1007/s10618-024-01016-z

[3] Explanations Go Linear: Post-hoc Explainability for Tabular Data with Interpretable Meta-Encoding, https://arxiv.org/pdf/2504.20667v2

[4] Black Box Explanation by Learning Image Exemplars in the Latent Feature Space, https://arxiv.org/pdf/2002.03746

[5] The Shapley Taylor Interaction Index, https://arxiv.org/pdf/1902.05622

[6] Explaining a series of models by propagating Shapley values, https://www.nature.com/articles/s41467-022-31384-3

**Audience:**

Yes

**Audience Explanation:**

Developing a reliable feature-attribution method applicable to general black-box, composite AI systems is of great practical relevance. In this regard, the paper should be of interest to researchers working in the field of feature attribution and even to the broader XAI research community.

**Claims And Evidence:**

Yes

**Claims Explanation:**

The paper provides a range of thoughtful experiments on the proposed method, showcasing its generality and effectiveness in attribution accuracy. Although the compared baselines are limited and somewhat outdated, I think the general critiques are valid, and the experimental results largely support the claims.

**Requested Changes:**

1. Add a dedicated related work section to systematize the state of the art in feature attribution methods. Consider adding a table to highlight the key differences among methods.

2. Reexamine the statements that criticize existing methods and the relevant literature. Use more cautious language when appropriate.

3. Benchmark the performance of SensX with more recent baselines.

4. Separate the long Section 2 using subsections. Having 7 short sections makes the flow difficult to follow.

---

> ### Author Response · Authors · 2026-04-03
>
> **Requested change 1: Add a dedicated related work section with a comparison table.**
>
> We have added a dedicated Related Work section (Section 2) covering Shapley-based methods (KernelSHAP, DeepSHAP, DeepLIFT, LRP, G-DeepSHAP, LIME, Shapley–Taylor interaction index, ILLUME), gradient-based methods (Integrated Gradients, Grad-CAM), perturbation and masking methods (RISE), learning-to-explain methods (L2X, INVASE), attention-based methods (attention rollout, Chefer et al. 2021), and Sobol-based methods. A comparison table summarizes all methods across three dimensions: background/reference requirement, model-agnosticism, and per-feature resolution.
>
> **Requested change 2: More cautious language about existing methods.**
>
> We have revised the language throughout. The introduction now explicitly acknowledges that workarounds exist: *"While workarounds exist for individual limitations (e.g., differentiable surrogate modeling for gradient access, dataset-based baselines for reference selection, or Shapley-value propagation through model chains), each introduces additional assumptions that may not hold in practice."*
>
> The specific references cited are addressed as follows.
>
> - **Refs [1], [2], [5], and [6]:** covered in Related Work, Shapley-based paragraph. See response to Requested change 3.
> - **Ref [3]** (surrogate modeling for IG access): we acknowledge this workaround in the Introduction and note that it introduces surrogate fidelity assumptions. Any such benchmark would conflate IG's attribution quality with surrogate fidelity, making the result uninterpretable in the composite settings where SensX is most relevant.
> - **Ref [4]** (Guidotti et al. 2019, ABELE): this method is constrained to image classification by design because its neighborhood generation and explanation extraction depend on an adversarial autoencoder trained on image data. Hence, we did not add this as a baseline.
>
> **Requested change 3: Benchmark against more recent baselines.**
>
> Our KernelSHAP (SHAP v0.48, official Python release March 2026) results use the antithetic (paired) sampling variant of Covert and Lee (2021), which achieves near-best convergence among model-agnostic estimators.
>
> *Added baseline to Section 4.1:* We have added PermutationExplainer (Covert and Lee 2021) at the same evaluation budget as KernelSHAP on the synthetic benchmarks. SensX outperforms both Shapley estimators, confirming that the accuracy gap is attributable to the feature removal assumption rather than estimator choice.
>
> - **Refs [1] and [2]** catalogue advances in Shapley estimators — methods that converge faster or with lower variance to the same limiting Shapley values. KernelSHAP with antithetic sampling still required approximately 33 TB for the ViT and 230 GB for the single-cell classifiers, and variance reduction does not change the fundamental memory scaling with feature count.
> - **Ref [5]** (Shapley–Taylor Interaction Index, Dhamdhere et al. 2020): We did not add it as a benchmark for three reasons. (1) At $k=1$ it reduces exactly to the standard Shapley value already estimated by KernelSHAP. (2) For $k>1$ the exact computation is exponential in the number of features, making it less tractable than KernelSHAP at the feature counts in our case studies. (3) It still requires a feature removal definition, inheriting the same reference dependency.
> - **Ref [6]** (G-DeepSHAP, Chen et al. 2022): G-DeepSHAP is not model-agnostic, requires each module to be a supported type with exposed internals, requires a shared background distribution at each interface, and cannot be applied when any module is a frozen encoder or API-only component. We do not claim it is never applicable; we claim it is inapplicable in the specific composite system that we analyzed.
> - **Conditional SHAP** (Aas et al. 2021): recovers feature importance conditional on training-data co-occurrence rather than isolated model dependence. We do not benchmark it for three reasons. (1) The synthetic benchmarks use independent Gaussian features, so conditional and marginal SHAP converge to the same values and the comparison would be uninformative. (2) Conditional SHAP requires estimating conditional distributions over the full feature space, which is at least as expensive as KernelSHAP and adds the cost of fitting a conditional model — infeasible at 150K features (ViT) or 27K features (single-cell). (3) It answers a different question from SensX, not a harder version of the same question. This distinction is discussed in Related Work and in the Manifold objection paragraph of the Discussion.
>
> **Requested change 4: Separate the long SensX section using subsections.**
>
> Restructured the SensX section from 7 paragraphs to 4 subsections: local neighborhood and characteristic perturbation scale, SensX values, theoretical characterization of SensX values, and implementation details.

---

> > ### Comment · Action_Editor_SaGp · 2026-04-13
> >
> > Dear Reviewer,
> > Could you please have a look at the authors' response and update your review accordingly ?
> >
> > Best
> > Amartya

---

### Author Response · Authors · 2026-04-21

We'd like to thank all reviewers for their timely constructive feedback and questions. We’ve carefully addressed each point in our responses and have revised the paper draft accordingly. All updated sections are highlighted in blue for clarity.

We look forward to continued discussion and are happy to address any further questions or suggestions.

---

### Decision · Action_Editor_SaGp · 2026-04-23

**Recommendation:** Accept as is

**Audience:**

Yes

**Audience Explanation:**

Feature attribution is of wide interest to a range of sub-communities within the TMLR audience.

**Claims And Evidence:**

Yes

**Claims Explanation:**

The authors tested SensX across four distinct, highly relevant case studies: synthetic benchmarks, Vision Transformers (ViTs), single-cell data, and composite spatial transcriptomics pipelines. Across these very diverse scenarios, the empirical results consistently show that that SensX outperforms established baselines like KernelSHAP and Integrated Gradients, particularly in black-box settings.The authors have also clearly defined the boundaries of their claims, especially after the review.  The method's theoretical limitations (e.g., behavior with general non-additive models at finite scales) are documented and the authors updated their claims regarding existing baselines to use more cautious, precise language. So overall, the claims are well supported.